



# The Multi-Scale Coupled Model: a New Framework Capturing Wind Farm-Atmosphere Interaction and Global Blockage Effects

Sebastiano Stipa[1], Arjun Ajay[1], Dries Allaerts[2], and Joshua Brinkerhoff[1]

[1]University of British Columbia, Okanagan Campus, CA
[2]Delft University of Technology, NL

**Correspondence:** Sebastiano Stipa (sebstipa@mail.ubc.ca)

**Abstract.** The growth in the number and size of wind energy projects in the last decade has revealed structural limitations in the current approach adopted by the wind industry to assess potential wind farm sites. These limitations are the result of neglecting the mutual interaction of large wind farms and the thermally-stratified atmospheric boundary layer. While currently available analytical models are sufficiently accurate to conduct site assessments for isolated rotors or small wind turbine clusters, the wind farm's interaction with the atmosphere cannot be neglected for large-size arrays. Specifically, the wind farm displaces the boundary layer vertically, triggering atmospheric gravity waves that induce large-scale horizontal pressure gradients. These perturbations in pressure alter the velocity field at the turbine locations, ultimately affecting global wind farm power production. The implication of such dynamics can also produce an extended blockage region upstream of the first turbines and a favorable pressure gradient inside the wind farm. In this paper, we present the multi-scale coupled (MSC) model, a novel approach that allows the simultaneous prediction of micro-scale effects occurring at the wind turbine scale, such as individual wake interactions and rotor induction, and meso-scale phenomena occurring at the wind farm scale and larger, such as atmospheric gravity waves. This is achieved by evaluating wake models on a spatially-heterogeneous background velocity field obtained from a reduced-order meso-scale model. The MSC model is validated against two large-eddy simulations (LES) with similar average inflow velocity profiles and a different capping inversion strength, so that two distinct interfacial gravity wave regimes are produced, i.e. subcritical and supercritical. Interfacial waves can produce high blockage in the first case, as they are allowed to propagate upstream. Conversely, in the supercritical regime their propagation speed is less than their advection velocity and upstream blockage is only operated by internal waves. The MSC model not only proves to successfully capture both local induction and global blockage effects in the two regimes, but also captures wind farm gravity-wave interaction, underestimating wind farm power by about only 2% compared with the LES results. Conversely, wake models alone, even if combined with a local induction model, cannot distinguish between differences in thermal stratification, and are affected by a first-row over-prediction bias that leads to a consistent overestimation of the wind farm power by 13% to 20%.

## 1 Introduction

The growth in the number and size of wind energy projects in the last decade has revealed structural limitations in the current approach adopted by the wind industry to assess potential wind farm sites. According to updates published by Ørsted (2019)

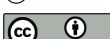



on its long-term financial targets, the existing engineering models used for wind farm site assessments are unable to accurately represent two important physical aspects that affect wind farm power production forecasts, namely, wake and blockage effects. Wake effects are the interaction of individual turbine wakes as they merge to form an extended region of reduced momentum downstream of the wind farm. Such a process is characterized by complex flow dynamics, and cannot be described by a simple combination of individual wake deficits. In fact, cluster wake formation is driven by additional phenomena, such as the effect that the increased shear stress has on vertical momentum fluxes, the effect that the latter have on the wake itself, and the formation of an internal boundary layer (IBL) that grows from the wind farm start. In addition, since more plants are being constructed in the proximity of pre-existing ones (Nygaard et al., 2020), cluster wake evolution is of great interest in the context of farm-farm wake interactions. Regarding blockage, or induction (Bleeg et al., 2018), this is defined as the flow deceleration upstream of the wind farm produced by the combination of individual (local) blockage created by individual turbines with a bulk (global) deceleration caused by the wind farm interacting with the thermally stratified atmosphere.

Many years of research on turbine-level induction resulted in different local-blockage models, such as the empirical formulation of Troldborg and Meyer Forsting (2017), the Rankine half-body of Gribben and Hawkes (2019), the vortex cylinder model (Branlard and Gaunaa, 2014; Branlard and Meyer Forsting, 2020), the self-similar model of Branlard et al. (2020) or the linearized model by Segalini (2021). Initially, global blockage effects produced by the entire cluster have been assessed by linearly superimposing local blockage models. As long as thermal stratification of the Earth's atmosphere was not considered, this superposition compared well against numerical simulations (see for example Branlard and Meyer Forsting, 2020). However, once atmospheric stability is considered, Centurelli et al. (2021) showed that linearly combining individual effects underpredicts the global blockage, and the inaccuracy worsens for larger wind farms. On the other hand, several numerical studies provided evidence that global blockage is a complex physical phenomenon that mainly depends on the structure of the potential temperature profile (Wu and Porté-Agel, 2017; Allaerts and Meyers, 2017, 2018). Specifically, upstream wind slowdown is produced by large-scale pressure perturbations that arise as a consequence of interfacial waves in the inversion layer and internal gravity waves aloft. To model these effects, Allaerts and Meyers (2019) developed the three-layer model (3LM), enhancing the earlier two-layer model (2LM) by Smith (2010), which enabled them to capture wind farm/gravity wave interactions in a fast engineering model for the first time. The idea behind the 3LM is to provide a meso-scale velocity correction for the free stream wind speed used in conventional wake models (Katić et al., 1987; Ainslie, 1988; Larsen, 1988; Anderson, 2009; OpenWind, 2010; Bastankhah and Porté-Agel, 2014; Niayifar and Porté-Agel, 2016). Such a correction has the effect of reducing the free stream velocity, reflecting the influence of atmospheric gravity waves in decelerating the wind incident to the wind farm. However, despite the the important physical insights at the meso-scale level offered by the 3LM, the coupling between the turbine-scale wake effects and the meso-scale global effects is weak, such that the model fails to adequately transfer the influence of the large-scale physical processes to the flow at the turbine scale.

Although it has been used to enhance model performance in capturing wake effects rather than blockage, the idea of coupling two models operating at different scales was initially introduced by Frandsen et al. (2006). Later, Stevens et al. (2014) introduced the coupled wake boundary layer model (CWBL), where the top-down model of Calaf et al. (2010) (later extended by Meneveau, 2011 and Stevens, 2016) was coupled with a wake model to enhance wind farm power predictions, assuming



that the wind farm internal boundary layer (IBL) is fully-developed. Further improvements of the CWBL model were made by Shapiro et al. (2019a), and later by Starke et al. (2021), in what is currently referred to as the area-localized coupled (ALC) model, extending its applicability to those regions where the IBL is not fully developed. The coupling idea has also been used by Nishino and Dunstan (2020) to develop a two-scale momentum theory, which couples conventional momentum theory with the large-scale momentum balance above the wind farm layer.

Except for the 2LM and 3LM, all coupled models mentioned above do not aim to capture the meso-scale wind farm blockage effects. In particular, they do not account for thermal stratification above the atmospheric boundary layer (ABL), with the result that meso-scale stability and its effects on blockage remain a major source of uncertainty in wind farm power predictions. To fill this gap, we introduce the multi-scale coupled (MSC) model, a framework that allows one to couple the 3LM with any wake model while also including local-induction effects. The MSC overcomes the above-mentioned deficiency characterizing the original 3LM and its coupling technique, establishing a new modular framework in which additional meso-scale effects can be easily added in the future. In the present study, we validate the MSC against large eddy simulations (LES) of a wind farm consisting of 100 NREL 5-MW turbines in an aligned configuration, under subcritical and supercritical interface wave regimes. These conditions are obtained by adjusting the so-called inversion Froude number $F_r = U_b/\sqrt{g'H}$, where $U_b$ is the bulk velocity inside the boundary layer, $g'$ is the reduced gravity and $H$ is the inversion height (Smith, 2010). In the subcritical regime, characterized by $F_r < 1$, interface waves exhibit a higher group velocity than the advection speed, allowing them to propagate upstream. This leads to higher experienced blockage than supercritical conditions ($F_r > 1$), where upstream perturbations are mainly due to internal waves. To highlight the improved accuracy of the MSC model, the LES results are also compared against the wake-only approach, with and without local induction, and to the original 3LM formulation by Allaerts and Meyers (2019).

The present paper is organized as follows. Section 2 briefly introduces the original 3LM model, followed by a detailed description of the proposed MSC model framework. Then, Section 3 presents the high-fidelity LES data that will be used for validation. Section 4 focuses on evaluating the predictive accuracy of the proposed engineering model, from the viewpoint of both the turbine data and the mean velocity field surrounding the wind farm. Finally, Section 5 highlights the conclusions of the present study.

## 2 Methodology

The multi-scale coupled (MSC) model describes the interaction between physical processes that take place inside the atmospheric boundary layer, such as turbine wakes or local rotor induction, and phenomena that exist aloft, characterized by larger spatial scales, such as wind farm-gravity wave interaction. The micro-scale to meso-scale coupling is deemed possible by introducing the notion of a background velocity field, which represents the effect of gravity wave-induced pressure gradients on an initially uniform velocity field. This background wind is then combined with wake and local induction effects, allowing the MSC model to capture not only the latter physical processes, but also additional phenomena such as global blockage effects, the favorable pressure gradient inside the wind farm, and large-scale velocity oscillations in the wind farm wake for those con-





ditions where gravity wave trains induce alternating favorable-unfavorable pressure gradients past the wind farm. The MSC model is formulated as a modular combination of different sub-models, which are suitably integrated and made interdependent. 95 This approach does not require additional tuning parameters or individual sub-model re-tuning.

Although the MSC framework generally differs from the original 3LM developed by Allaerts and Meyers (2019), especially in the solution strategy and sub-model coupling, the two models share some common aspects, such as the chosen wake model or the meso-scale modeling approach. For this reason, we first introduce the original 3LM formulation in Section 2.1, together with all common aspects between the two models. Later, in Section 2.2, we present the new methodology, focusing on its 100 differences and improvements with respect to the original formulation.

## 2.1 The Three Layer Model

Developed by Allaerts and Meyers (2019), the three layer model (3LM) aims at capturing wind farm gravity wave interaction by combining a micro and a meso-scale model. The well-known Bastankhah and Porté-Agel (2014) wake model is used at the micro-scale level, while a substantially improved evolution of the two layer model (2LM) of Smith (2010) is adopted to 105 model meso-scale effects produced by gravity waves. Although very promising due to the number of physical processes that it can capture, the 3LM suffers from two limitations. Firstly, the fact that it solves for depth-averaged velocities complicates its coupling with wake models, which are usually formulated in terms of the actual wind speed. Secondly, the coupling method only transfers the gross wind farm blockage at the micro-scale level, omitting the effect of gravity waves inside the wind farm and in its wake. The present section is organized as follows. Section 2.1.1 explains the meso-scale model used within the 110 3LM, Section 2.1.2 outlines the adopted wake model, Section 2.1.3 describes the turbulence intensity (TI) model that controls wake expansion within the wake model, while Section 2.1.4 explains the wake superposition strategy adopted in the original 3LM formulation. Finally, Section 2.1.5 outlines the technique used by Allaerts and Meyers (2019) to couple each sub-model, highlighting its issues and limitations.

### 2.1.1 Meso-Scale Model

115 The 3LM exploits the theory for interacting gravity waves and boundary layers developed by Smith (1980, 2007, 2010), with extra features such as the Coriolis force, the additional wind farm layer that relaxes Smith's homogeneous vertical mixing assumption, and the wind farm gravity wave coupling mechanism. Specifically, the model is formulated by dividing the vertical ABL structure into three layers, i.e. the wind farm layer, the upper layer, and the free atmosphere or geostrophic layer aloft (a conceptual sketch is given in Figure 1).





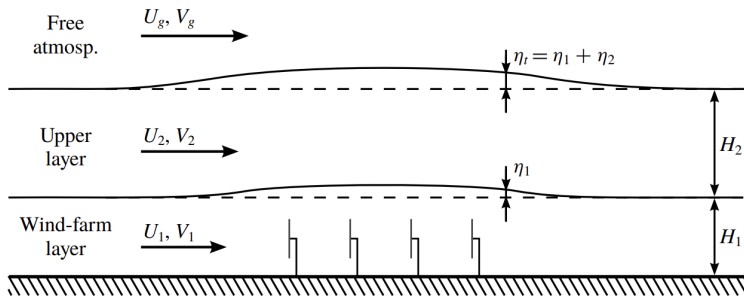

**Figure 1.** Conceptual sketch of the three layer model, reprinted with permission from Allaerts and Meyers (2019).

The depth-averaged Navier-Stokes equations used in the first two layers, linearised around a reference ABL state, are

$$
\begin{cases}
U_1 \dfrac{\partial u_1}{\partial x} + V_1 \dfrac{\partial u_1}{\partial y} + \dfrac{1}{\rho}\dfrac{\partial p}{\partial x} = -f_c v_1 + \nu_{t,1}\left(\dfrac{\partial^2 u_1}{\partial x^2} + \dfrac{\partial^2 u_1}{\partial y^2}\right) + \dfrac{D_{11}}{H_1}(u_2 - u_1) + \dfrac{D_{12}}{H_1}(v_2 - v_1) - \dfrac{C_{11}}{H_1}u_1 - \dfrac{C_{12}}{H_1}v_1 - \dfrac{f_x}{H_1} \\[2mm]
U_1 \dfrac{\partial v_1}{\partial x} + V_1 \dfrac{\partial v_1}{\partial y} + \dfrac{1}{\rho}\dfrac{\partial p}{\partial y} = f_c u_1 + \nu_{t,1}\left(\dfrac{\partial^2 v_1}{\partial x^2} + \dfrac{\partial^2 v_1}{\partial y^2}\right) + \dfrac{D_{21}}{H_1}(u_2 - u_1) + \dfrac{D_{22}}{H_1}(v_2 - v_1) - \dfrac{C_{21}}{H_1}u_1 - \dfrac{C_{22}}{H_1}v_1 - \dfrac{f_y}{H_1} \\[2mm]
U_1 \dfrac{\partial \eta_1}{\partial x} + V_1 \dfrac{\partial \eta_1}{\partial y} + H_1\left(\dfrac{\partial u_1}{\partial x} + \dfrac{\partial v_1}{\partial y}\right) = 0
\end{cases}
\tag{1}
$$

and

$$
\begin{cases}
U_2 \dfrac{\partial u_2}{\partial x} + V_2 \dfrac{\partial u_2}{\partial y} + \dfrac{1}{\rho}\dfrac{\partial p}{\partial x} = -f_c v_2 + \nu_{t,2}\left(\dfrac{\partial^2 u_2}{\partial x^2} + \dfrac{\partial^2 u_2}{\partial y^2}\right) - \dfrac{D_{11}}{H_2}(u_2 - u_1) - \dfrac{D_{12}}{H_2}(v_2 - v_1) \\[2mm]
U_2 \dfrac{\partial v_2}{\partial x} + V_2 \dfrac{\partial v_2}{\partial y} + \dfrac{1}{\rho}\dfrac{\partial p}{\partial y} = f_c u_2 + \nu_{t,2}\left(\dfrac{\partial^2 v_2}{\partial x^2} + \dfrac{\partial^2 v_2}{\partial y^2}\right) - \dfrac{D_{21}}{H_2}(u_2 - u_1) - \dfrac{D_{22}}{H_2}(v_2 - v_1) \\[2mm]
U_2 \dfrac{\partial \eta_2}{\partial x} + V_2 \dfrac{\partial \eta_2}{\partial y} + H_2\left(\dfrac{\partial u_2}{\partial x} + \dfrac{\partial v_2}{\partial y}\right) = 0,
\end{cases}
\tag{2}
$$

where subscripts 1 and 2 indicate the wind farm and upper layer respectively, $\eta$ is the layer displacement, $H$ is the layer height, $\mathbf{U} = (U, V)$ and $\mathbf{u} = (u, v)$ are the depth-averaged background and perturbation velocities respectively, $\nu_t$ is the eddy viscosity (see Section 2.3), and $f_c = 2\omega_z$ is the Coriolis parameter, with $\omega_z$ the component of the Earth's rotation rate vector normal to the sea-level geopotential surface approximated as a sphere (Gill, 1982). The term $\mathbf{f}/H_1$ in Equation (1), is the wind farm forcing term and enables the wind farm/gravity wave feedback mechanism (see Section 2.1.5 for the definition of this term). Tensors $C_{ij}$ and $D_{ij}$ describe the perturbation of friction at the ground and at the top of the wind farm layer respectively, defined as

$$
C_{ij} = \frac{\left\|\tau|_{z=0}\right\|}{\left\|\mathbf{U}_1\right\|^3}
\begin{bmatrix}
2U_1^2 + V_1^2 & U_1 V_1 \\
V_1 U_1 & U_1^2 + 2V_1^2
\end{bmatrix}
\tag{3}
$$





and

$$D_{ij} = \frac{\left\|\tau|_{z=H_1}\right\|}{\|\Delta\mathbf{U}\|^3} \begin{bmatrix} 2\Delta U^2 + \Delta V^2 & \Delta U \Delta V \\ \Delta V \Delta U & \Delta U^2 + 2\Delta V^2 \end{bmatrix}, \tag{4}$$

where $\Delta\mathbf{U} = (\Delta U, \Delta V) = (U_2 - U_1, V_2 - V_1)$ is the velocity difference, component by component, between the upper and
wind farm layer, while $\|\cdot\|$ denotes L2 vector norm.

The third layer, which is effectively a boundary condition, relates the total ABL vertical displacement to the perturbation
pressure felt inside the ABL by

$$\frac{1}{\rho_0}\hat{p} = \left[\frac{i\left(N^2 - \Omega^2\right)}{m} + g'\right]\hat{\eta} \tag{5}$$

based on the linear, three-dimensional, non-hydrostatic, non-rotating gravity wave theory of Smith (1980) (see also Lin, 2007
and Nappo, 2012). The top-hat symbol is used to indicate that Equation (5) is expressed in Fourier space, $\eta = \eta_1 + \eta_2$ is the
total vertical ABL displacement, $N = \sqrt{\gamma g/\theta_0}$ is the Brunt-Väisälä frequency, where the lapse rate $\gamma = d\theta/dz$ is assumed
constant in this study, $g' = g\Delta\theta/\theta_0$ is referred to as the reduced gravity, where $\Delta\theta$ is the potential temperature jump across
the inversion layer and $\theta_0$ is the reference potential temperature. The quantity $m$ identifies the vertical wavenumber, evaluated
using the dispersion relation

$$m^2 = (k^2 + l^2)\left(\frac{N^2}{\Omega^2} - 1\right) \tag{6}$$

in which $\boldsymbol{\kappa} = (k, l)$ is the horizontal wavenumber and $\Omega = -\boldsymbol{\kappa} \cdot \mathbf{U}_3$ is the intrinsic wave frequency.

After determining the background state (see Section 2.3), Equations (1), (2) and (5) are numerically solved on a 2D Cartesian
grid using a Fourier-Galerkin discretization to yield the perturbation velocity $\mathbf{u}$ and displacement $\eta$ in the wind farm and upper
layers, as well as the perturbation pressure $p$. It is important to highlight the significance of the pressure variable, as this
concept will be essential in the formulation of the MSC framework later on. First of all, Allaerts and Meyers (2019) make
the assumption of zero vertical pressure gradient inside the ABL, hence $p$ only varies in the horizontal directions. Secondly,
looking at Equation (5), it is clear that $p$ can only change in response to a vertical displacement of the ABL and only if the
latter experiences a capping inversion jump $\Delta\theta$ and/or a stable lapse rate $\gamma$. For this reason, the pressure variable considered
by Allaerts and Meyers (2019) only contains the effect of internal and interfacial waves, and does not account for the local
pressure rise in front of each wind turbine, responsible for what we refer to as local blockage or turbine-level induction.

### 2.1.2 Wake Model

The 3LM has to be complemented with a wake model in order to predict wind turbine thrust and power. In this regard, for
each wind turbine, we start by considering a wake coordinate system $\mathbf{e}_k$ having its origin at the turbine rotor center, the $x$ axis
aligned with the local free stream velocity, the $z$ axis directed as the turbine tower from base to top, and the $y$ axis to form a
right-handed frame of reference. Bastankhah and Porté-Agel (2014) have shown that the non-dimensional velocity deficit at a





point $P(\mathbf{x})$ in the wind turbine wake assumes a self-similar profile, and can be expressed as

$$\frac{U_\infty - U(\mathbf{x})}{U_\infty} = W(\mathbf{x}) = C(x) f\big(r(\mathbf{x}), \sigma(x)\big) \mathcal{H}(x) \tag{7}$$

where $U_\infty$ is the free stream velocity magnitude, $C(x)$ is the maximum normalized velocity deficit, $r(\mathbf{x})$ is the radial distance
between $P$ and the wake centerline, $\sigma(x)$ is a length scale related to the wake width, which increases with increasing down-
165   stream distance from the rotor, $f(r, \sigma)$ is a similarity function describing the shape of the velocity deficit profile and $\mathcal{H}$ is the
Heaviside function. Different expressions exist for $C(x)$ and $f(r, \sigma)$ (see Lanzilao and Meyers, 2022a for a review), which
identify the specific wake model.

In this study, following Allaerts and Meyers (2019), we use the formulas developed by Bastankhah and Porté-Agel (2014),
corresponding to the Gaussian wake model, which evaluate $f(r, \sigma)$ and $C(x)$ as

170   $$f(r, \sigma) = \exp\left(-\frac{r^2}{2\sigma^2}\right) \tag{8}$$

$$C(x) = 1 - \sqrt{1 - \frac{C_T}{8(\sigma/D)^2}} \tag{9}$$

where $C_T$ and $D$ are the turbine thrust coefficient and diameter respectively. A known problem of the Gaussian wake model
is that it doesn't conserve momentum when $C_T > 8\sigma^2/D^2$. Specifically, depending on the value of the $C_T$ coefficient, Equa-
tion (9) can be undefined up to a certain downstream distance from the wind turbine. Such limit in the model formulation can
175   be disregarded if Equation (7) is evaluated at wind turbine locations only, as real-life streamwise spacing generally satisfies
$C_T < 8\sigma^2/D^2$. Conversely, if one aims at predicting the entire flow field using the wake model, a correction in the near wake
region is needed to avoid deficit saturation at high $C_T$ values. In the present study, we deal with this issue by using a damping
function that gradually shifts from the Gaussian to the super-Gaussian wake model in the near wake (see Appendix A for
details). The resulting wake model is fully equivalent to the Gaussian model after 4 downstream diameters, but it allows to
180   compute the velocity in the near wake by removing the singularity in Equation (9). Similarly to Jensen (1983), Bastankhah and
Porté-Agel (2014) express the wake width $\sigma$ as a linear function of $x$, namely

$$\sigma(x) = k^* x + c_s \sqrt{\beta} D \tag{10}$$

$$\beta = \frac{1}{2} \frac{1 + \sqrt{1 - C_T}}{\sqrt{1 - C_T}} \tag{11}$$

$$k^* = a_s TI + b_s \tag{12}$$

185   where, according to Niayifar and Porté-Agel (2016), $a_s = 0.3837$, $b_s = 0.003678$ and $c_s = 0.2$. The TI value is evaluated at
the location of the turbine which is shedding the wake, and its calculation is detailed in Section 2.1.3.

### 2.1.3 Turbulence Intensity Model

A key parameter of any wake model is the wake expansion coefficient $k^*$. Equation (12) shows that this parameter is related
to the TI level at the wind turbine that is shedding the wake. This normally varies throughout the wind farm, with the strongest





variation happening between the first row, where the background ambient TI is experienced, and the downstream rows, where the TI level is increased due to enhanced mixing operated by the wakes. In the present study, we adopted the TI model proposed by Niayifar and Porté-Agel (2016), where the actual turbulence intensity at each turbine location is given by

$$TI_k = \sqrt{TI_\infty^2 + TI_{+,k}^2},$$

where $TI_\infty$ is the background turbulence intensity provided as a user-defined input, while $TI_{+,k}$ is the added turbulence intensity, evaluated as

$$TI_{+,k} = \max_j \left( \mathcal{H}(x_k|_j) I_j(x_k|_j) \frac{A_w}{\pi R_k^2} \right) \tag{13}$$

where $\mathcal{H}$ is again the Heaviside function, $x_k|_j$ is the distance between turbine $k$ and turbine $j$ rotor center, in the wake reference frame of turbine $j$, $R_k$ is rotor radius of turbine $k$, $A_w$ is an area which value goes from $0$ to $\pi R_k^2$, depending on whether turbine $j$ is waking $k$ or not, with $0$ corresponding to the non-waking situation (see Niayifar and Porté-Agel, 2016). The rationale behind Equation (13) is that the value of $TI_+$ at turbine $k$ is the maximum among all TI values produced by other turbines, as if they were waking turbine $k$ individually. The parameter $I_j$ is an empirical function due to Crespo and Hernandez (1996) which is given by

$$I_j(x) = d_s \left( \frac{1}{2}(1 - \sqrt{1 - C_{T,j}}) \right)^{e_s} TI_\infty^{f_s} \left( \frac{x}{D_j} \right)^{g_s} \tag{14}$$

in the present study, we use the values proposed by Niayifar and Porté-Agel (2016) for $e_s$, $f_s$ and $g_s$, equal to $0.8325$, $0.0325$ and $-0.32$ respectively, while we vary $d_s$ in order to minimize $\left\| TI_{k,LES} - TI_{k,Mod} \right\|$, i.e. the L2 norm of the TI difference at each turbine location between our LES simulations (Section 3.3) and the model. This leads to $d_s = 0.8798$ instead of the original value of $0.73$ proposed by Niayifar and Porté-Agel (2016). The reason why we decided to re-tune the TI model comes from the fact that Equation (14) starts to be valid after 5 downstream diameters from turbine $k$, which is exactly equal to the streamwise turbine spacing we chose in our LES simulations. This, combined with the choice to simulate an aligned wind farm, determined that the added turbulence intensity at each waked turbine is always produced by the turbine directly upstream, due to Equation (13). As a consequence, using the original $d_s$ coefficient, the TI model operates at the edge of its validity bounds inside the wind farm, which we observed to produce poor results (in particular, underestimating $TI_{+,k}$). Conversely, the proposed value of $d_s$ allows to run the wake model with a $TI$ value that closely matches the LES simulations at each turbine location. The reported coefficients for the turbulence intensity model have been used in all wake model runs performed throughout the present paper.

### 2.1.4 Wake Superposition

When dealing with large wind farm arrays, turbine wakes can overlap, resulting in full or partial waking of downstream turbines. For this reason, individual velocity deficits have to be combined by means of wake superposition methods. Two main





techniques have been extensively used,

$$U(\mathbf{x}) = U_\infty - \sum_{k=1}^{N_t} u_k W_k(\mathbf{x}) \qquad (15)$$

$$U(\mathbf{x}) = U_\infty - \sqrt{\sum_{k=1}^{N_t} \left( u_k W_k(\mathbf{x}) \right)^2} \qquad (16)$$

where $k$ denotes the $k$-th wind turbine, $N_t$ is the number of turbines in the array, $u_k$ is a velocity to be defined, associated with turbine $k$, and $U_\infty$ is the free stream velocity, assumed uniform for now, meaning that wake coordinate systems $\mathbf{e}_k$ are equal for all $k$. While Equation (15), first proposed by Lissaman (1979), consists in summing velocity deficits, Equation (16) sums the squares of velocity deficits, and was introduced by Katić et al. (1987). These studies made the additional assumption of $u_k = U_\infty$, leading to a simple summation of $W(\mathbf{x})$ or $W(\mathbf{x})^2$. This produces too high deficits, and has been addressed by Niayifar and Porté-Agel (2016) and Voutsinas et al. (1990), who used $u_k = U(\mathbf{x}_k)$ with $\mathbf{x}_k$ the location of the $k$-th wind turbine for Equations (15) and (16), respectively. This introduces some sort of weighting based on the velocity experienced at the turbine that is shedding the wake.

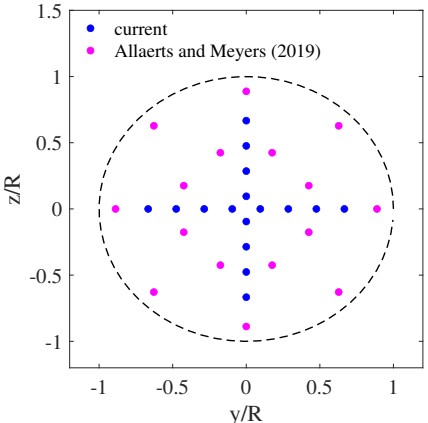

**Figure 2.** Comparison between the quadrature point distribution adopted in the present study and in Allaerts and Meyers (2019).

To be able to account for partial waking of downstream turbines, we follow the approach of Allaerts and Meyers (2019) and model each turbine using $N_q = 16$ quadrature points, evaluating $u_k$ as

$$u_k = \frac{1}{N_q} \sum_{q=1}^{N_q} U(\mathbf{x}_{k,q}) \qquad (17)$$

where $\mathbf{x}_{k,q}$ indicates the coordinates of the $q$-th quadrature point belonging to the $k$-th turbine. This greatly enhances the wake model performance in partial waking, or when a vertical velocity gradient is considered. In the current study, we use a cross-type quadrature points distribution rather than the star-type arrangement adopted by Allaerts and Meyers (2019) (see Figure 2





for a comparison between the two distributions). Despite the cross-type arrangement only features two azimuthal directions (horizontal and vertical), it allows for a finer resolution along the rotor radius with the same number of quadrature points. This is, in our opinion, more suited to capture the effects produced by vertical velocity gradients and partial waking conditions.

When presenting results from the original 3LM formulation of Allaerts and Meyers (2019), we use Equation (15) to combine
velocity deficits from the Bastankhah and Porté-Agel (2014) model. Conversely, the coupling methodology of the MSC model requires a different superposition method, which is described in Section 2.2.

### 2.1.5    Original Coupling Technique

In order to capture wind-farm gravity wave feedback, the meso-scale model and the wake model need to be suitably coupled. For conventional wake models, the thrust force associated with the $k$-th turbine ultimately depends on the unperturbed free
stream velocity, i.e. $\mathbf{f}_k = \mathbf{f}_k(U_\infty)$. In the coupling method proposed by Allaerts and Meyers (2019), $\mathbf{f}_k$ is corrected for blockage effects based on a perturbation velocity $u^{\mathrm{up}}$ so that $\mathbf{f}_k = \mathbf{f}_k(U)$, where $U = U_\infty + u^{\mathrm{up}}$. The point-wise perturbation velocity $u^{\mathrm{up}}$ is calculated by evaluating the meso-scale perturbation velocity $u_1(\mathbf{x})$ at $\mathbf{x}^{\mathrm{up}}$, a point located upstream the first wind farm row, which is deemed representative of the global blockage effect for the entire farm. For a free stream wind speed aligned with the $x$ direction, Allaerts and Meyers (2019) evaluated the coordinates of the upstream point as $x^{\mathrm{up}} = \min(x_k) - 10D$ and
$y^{\mathrm{up}} = 1/N_t \sum_{k=1}^{N_t} y_k$, where $x_k$ and $y_k$ are the rotor center coordinates of turbine $k$. Finally, $\mathbf{f}_k$ is filtered on the 3LM numerical mesh using a Gaussian kernel $G$. Such operation returns a smoothed wind farm body-force field, and can be expressed as

$$\mathbf{f}(x,y) = \int\limits_0^{L_x} \int\limits_0^{L_y} G(x-x', y-y') \sum_{k=1}^{Nt} \mathbf{f}_k \delta(x'-x_k, y'-y_k)\,dx'dy', \qquad (18)$$

with

$$G(x,y) = \frac{1}{\pi L^2} \exp\left(-\frac{x^2+y^2}{L^2}\right) \qquad (19)$$

where $\delta(x,y)$ is the Dirac's function and $L$ is the projection width, set to the greatest between the streamwise and spanwise meso-scale grid spacing.

Regarding the numerical algorithm used to solve the coupled problem, Allaerts and Meyers (2019) expressed $\mathbf{f}_k$'s dependency on $u^{\mathrm{up}}$ by expanding $\mathbf{f}_k(U)$ in Taylor series around $U_\infty$, truncating the polynomial at the first order. Such operation required the Jacobian of the wake model — specific to the chosen model — with respect to $u^{\mathrm{up}}$ to be derived. Moreover, it
produced a convolution term in the Fourier transform of Equation (1), yielding a system in Fourier space where wavenumbers are not mutually independent. In the present paper, we employ a slightly different but more flexible algorithm to solve the coupled problem that eliminates the need for the Jacobian computation. Specifically, we iterate between the meso-scale and the wake model, initializing the Gaussian-filtered thrust distribution by running the wake model with the uniform background wind $U_\infty$. This returns $u_k$, which is calculated at each turbine location using Equations (7) and (15). At this point, turbine
thrust is evaluated as

$$\mathbf{f}_k = \frac{1}{2} u_k^2 \frac{\pi D^2}{4} C_{T,k} \mathbf{e}_k \qquad (20)$$





where $\mathbf{e}_k$ is defined in Section 2.1.2 and $C_{T,k}$ is the thrust coefficient of the $k$-th turbine. After filtering $\mathbf{f}_k$ using Equation (18), the 3LM is solved using Equations (1), (2) and (5), and the free stream wind for the wake model is corrected as $U^{i+1} = U_\infty + u^{\mathrm{up}}$. The wake model is run again with the updated free stream wind by setting $U_\infty = U^{i+1}$ in Equations (7) and (15),

and the whole process is repeated until the relative residual between two consecutive iterations falls below a specified tolerance (we use the 3LM pressure to calculate the residual, i.e. $r = \left\| p^{i+1} - p^i \right\|_2 / \left\| p^{i+1} \right\|_2$, where $\| \|_2$ is the L2 norm). It is interesting to note that, in our approach, the Gaussian-filtered turbine thrust distribution resulting from the initialization step is equivalent to the the Gaussian-filtered 0-th order term of the Taylor series expansion proposed by Allaerts and Meyers (2019), while the iterative procedure removes the error initially represented by higher-order terms, correcting the entire thrust distribution at

each iteration. Moreover, in the system of equations resulting from our procedure each wavenumber is independent from the remaining ones, leading to a block-diagonal system matrix (see Appendix C).

Although the employed numerical algorithm is slightly different, we emphasize that the physical coupling between the meso-scale and the micro-scale is identical to the one proposed by Allaerts and Meyers (2019). Such coupling method is characterized by a non-local point-wise velocity correction that leads to the following structural limitations. First, the new free

stream velocity—although corrected for global blockage effects— is always the same for all turbines. Secondly, the upstream point location becomes ambiguous if an arbitrary wind farm geometry is considered, or if the wind farm contains multiple sizes of turbines. Finally, such coupling method completely disregards the beneficial effect of gravity wave-induced pressure gradients inside the wind farm.

## 2.2 The Multi Scale Coupled Model

In the present section, we introduce the multi-scale coupled (MSC) model, a framework and coupling technique aimed at modeling turbine-wake interactions, local turbine blockage, wind farm/gravity wave interactions, and global blockage effects. Specifically, this is achieved by combining different sub-models, and developing a suitable coupling technique that allows their inter-dependence. Strengths of the MSC model are its modularity and the fact that it builds upon already-existing sub-models, so that additional tuning parameters or individual sub-model re-tuning are not required. Ultimately, the MSC model aims to resolve

the structural deficiencies of the 3LM model, presenting itself as a novel modular framework to capture the interaction between micro- and meso-scale phenomena in wind farms in a stratified atmosphere. The present section is organized as follows. In the remainder we will introduce the MSC model, detailing the new coupling method in Section 2.2.1. Then, Section 2.2.2 explains the reconstruction technique used to transform the depth-averaged perturbation velocity produced by the meso-scale model into the actual background velocity field required by the wake model. Since the obtained background wind is not uniform in

space, we use a new wake superposition strategy, described in Section 2.2.3, that allows us to combine individual wake deficits with any heterogeneous background wind. Section 2.2.4 provides details on how local-induction effects are introduced, while Section 2.2.5 finally summarizes the MSC model solution procedure.

We start by defining a meso-scale and a micro-scale level, each associated with one or more respective sub-models. The meso-scale model is forced by the turbine thrust distribution, which is provided by the micro-scale model, while the latter is

forced by a heterogeneous change (both in magnitude and direction) of the background velocity experienced at each turbine





location. A conceptual sketch of this formulation is depicted in Figure 3. Two assumptions underlie this approach. Firstly, we assume that turbine and atmospheric processes occur at separate spatial scales. As a result, the micro-scale sub-model only accounts for effects at the turbine scale (such as wakes or local blockage), while the meso-scale model is responsible for modeling the gross background velocity change produced by physical processes happening at the wind farm and/or ABL

scale (such as wind farm/gravity wave interaction). Secondly, and most important, we assume that these effects can be linearly superimposed, i.e. the effective velocity field is given by the linear superposition of the mesoscale background wind speed and the wind speed changes predicted at the micro-scale level.

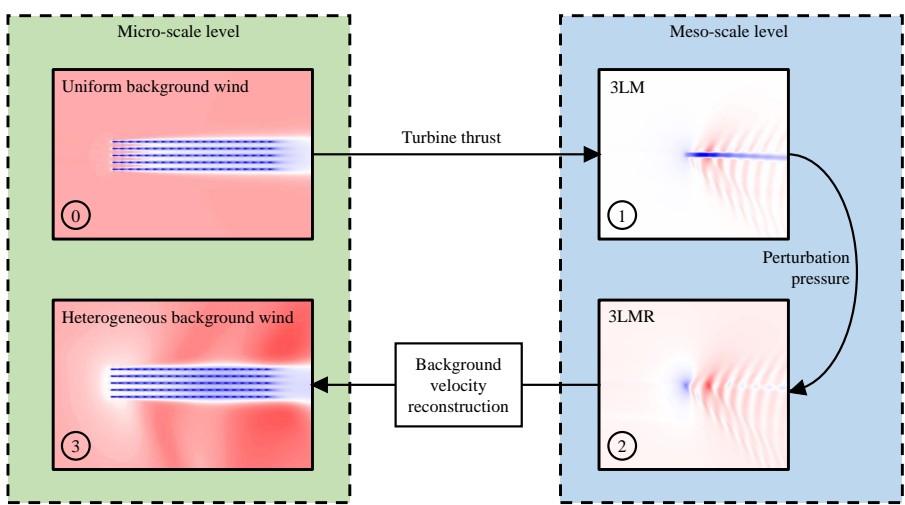

**Figure 3.** Conceptual sketch of the MSC model. After step 0, used to initialize turbine thrust distribution, steps 1, 2 and 3 are solved iteratively, updating turbine thrust and background velocity, until the residual on perturbation pressure has converged to a specified tolerance.

In the present study, we use a modified version of the 3LM to model meso-scale processes, while the Bastankhah and Porté-Agel (2014) wake model and the Branlard and Gaunaa (2014) vortex-cylinder models are used to capture micro-scale processes.

To couple the micro- and meso-scales without double-counting their effects, both the superposition and the separation-of-scales assumptions must hold. At the micro-scale level, wake models are usually tuned on velocity for an isolated wind turbine wake, hence, by construction, they do not account for turbine-ABL interaction. Also the vortex cylinder model, derived using a vorticity formulation, was developed to specifically account for turbine induction, and its results only depend on turbine characteristics, such as thrust coefficient, diameter, and free stream velocity at the turbine location. At the meso-scale level,

on the other hand, the perturbation velocity predicted by the original 3LM contains both gravity wave and wake effects, as equations are forced with the turbine thrust distribution. As a result, the original 3LM does not satisfy the superposition and separation-of-scales assumptions, and needs to be suitably extended with an additional solution step. The objective is to exclude wake effects from the 3LM solution, isolating the result of gravity-wave induced pressure gradients. Specifically, we recall that 3LM pressure only accounts for atmospheric stratification effects. Hence, after solving the original 3LM, i.e. Equations (1),





(2) and (5) and obtaining $p$, we solve an additional modified set of equations, where perturbation pressure $p$ is the forcing term and turbine thrust is removed. We denote such procedure as the "three-layer model reconstruction" (3LMR) step, performed inside the wind farm and upper layer. The result is a so-called background perturbation velocity that only contains the effect of large-scale pressure gradients produced by gravity waves.

### 2.2.1 Three Layer Model Reconstruction

Within the three-layer model reconstruction (3LMR) step, we use the perturbation pressure field obtained from the original 3LM to force a modified set of three-layer equations. It is important to note that, by fixing $p$, we are implicitly fixing the boundary layer displacement $\eta$ due to Equation (5). Hence, Equation (5) is automatically satisfied for the third layer, and only the momentum equations in the wind farm and upper layers are retained, reading

$$
\begin{cases}
U_1 \dfrac{\partial u_1^{\text{bk}}}{\partial x} + V_1 \dfrac{\partial u_1^{\text{bk}}}{\partial y} = -f_c v_1^{\text{bk}} + \nu_{t,1} \left( \dfrac{\partial^2 u_1^{\text{bk}}}{\partial x^2} + \dfrac{\partial^2 u_1^{\text{bk}}}{\partial y^2} \right) + \dfrac{D_{11}}{H_1}(u_2^{\text{bk}} - u_1^{\text{bk}}) + \dfrac{D_{12}}{H_1}(v_2^{\text{bk}} - v_1^{\text{bk}}) - \dfrac{C_{11}}{H_1} u_1^{\text{bk}} - \dfrac{C_{12}}{H_1} v_1^{\text{bk}} - \dfrac{1}{\rho}\dfrac{\partial p}{\partial x} \\[4mm]
U_1 \dfrac{\partial v_1^{\text{bk}}}{\partial x} + V_1 \dfrac{\partial v_1^{\text{bk}}}{\partial y} = f_c u_1^{\text{bk}} + \nu_{t,1} \left( \dfrac{\partial^2 v_1^{\text{bk}}}{\partial x^2} + \dfrac{\partial^2 v_1^{\text{bk}}}{\partial y^2} \right) + \dfrac{D_{21}}{H_1}(u_2^{\text{bk}} - u_1^{\text{bk}}) + \dfrac{D_{22}}{H_1}(v_2^{\text{bk}} - v_1^{\text{bk}}) - \dfrac{C_{21}}{H_1} u_1^{\text{bk}} - \dfrac{C_{22}}{H_1} v_1^{\text{bk}} - \dfrac{1}{\rho}\dfrac{\partial p}{\partial y}
\end{cases}
\tag{21}
$$


$$
\begin{cases}
U_2 \dfrac{\partial u_2^{\text{bk}}}{\partial x} + V_2 \dfrac{\partial u_2^{\text{bk}}}{\partial y} = -f_c v_2^{\text{bk}} + \nu_{t,2} \left( \dfrac{\partial^2 u_2^{\text{bk}}}{\partial x^2} + \dfrac{\partial^2 u_2^{\text{bk}}}{\partial y^2} \right) - \dfrac{D_{11}}{H_2}(u_2^{\text{bk}} - u_1^{\text{bk}}) - \dfrac{D_{12}}{H_2}(v_2^{\text{bk}} - v_1^{\text{bk}}) - \dfrac{1}{\rho}\dfrac{\partial p}{\partial x} \\[4mm]
U_2 \dfrac{\partial v_2^{\text{bk}}}{\partial x} + V_2 \dfrac{\partial v_2^{\text{bk}}}{\partial y} = f_c u_2^{\text{bk}} + \nu_{t,2} \left( \dfrac{\partial^2 v_2^{\text{bk}}}{\partial x^2} + \dfrac{\partial^2 v_2^{\text{bk}}}{\partial y^2} \right) - \dfrac{D_{21}}{H_2}(u_2^{\text{bk}} - u_1^{\text{bk}}) - \dfrac{D_{22}}{H_2}(v_2^{\text{bk}} - v_1^{\text{bk}}) - \dfrac{1}{\rho}\dfrac{\partial p}{\partial y}
\end{cases}
\tag{22}
$$

where subscripts 1 and 2 again refer to the wind farm and upper layers, respectively, and $\mathbf{u}^{\text{bk}} = (u^{\text{bk}}, v^{\text{bk}})$ is the background perturbation velocity.

We highlight that perturbation velocities resulting from Equations (21) and (22) only contain the effect of large-scale gravity
wave pressure gradients produced by the wind farm, while the direct influence of the latter is removed. However, since it is a depth-averaged perturbation quantity, $\mathbf{u}^{\text{bk}}$ cannot be directly used within the wake model. In the next section, we deal with its conversion from depth-averaged to an actual, height-dependent velocity field.

### 2.2.2 Background Velocity Reconstruction

In order to transform the depth-averaged perturbation field resulting from the 3LMR step into an actual velocity field — also
introducing height dependency — we developed a method that consistently matches the local background velocity profile $\mathbf{U_b}(\mathbf{x})$ with the layer average $\mathbf{U}_1 + \mathbf{u}_1^{\text{bk}}$ in the wind farm layer. Following Panofsky (1963), who used the similarity hypothesis of Monin and Obukhov (1954), the velocity profile in the ABL can be expressed as

$$
\|\mathbf{U_b}(z)\| = \frac{u^*}{\kappa} \left( \ln\left(\frac{z}{z_0}\right) - \Psi\left(\frac{z}{L}\right) \right)
\tag{23}
$$



where $u^*$ is the friction velocity, $z_0$ is the equivalent roughness length, $\kappa$ is the von Karman constant (which we set to $0.4$ in the present study), and $\Psi$ is a function of the Obukhov length scale $L$. For neutral stratification below the ABL height, $\Psi \to 0$, while for stable or unstable stratification different methods have been proposed to express $\Psi$, such as the method of Businger (1966) and Dyer (unpublished) for unstable conditions, or Etling (1996)'s method for the stable ABL. In the context of this study, we only consider conventionally neutral ABLs, so we set $\Psi = 0$. If the background flow exhibits small variations due to the effect of large-scale pressure gradients, it is reasonable to assume that the friction velocity $u^*$ varies horizontally, while the profile shape is preserved. In the present study, we evaluate $u^*(x,y)$ by matching the average velocity magnitude calculated exploiting Equation (23) with results obtained from the 3LMR, namely

$$\sqrt{\left(U_1 + u_1^{\text{bk}}\right)^2 + \left(V_1 + v_1^{\text{bk}}\right)^2} = \frac{u^*(x,y)}{\kappa(H_1 - z_0)} \int\limits_{z_0}^{H_1} \left(\ln\left(\frac{z}{z_0}\right)\right) dz, \tag{24}$$

which yields

$$u^*(x,y) = \frac{\kappa(H_1 - z_0)\sqrt{\left(U_1 + u_1^{\text{bk}}\right)^2 + \left(V_1 + v_1^{\text{bk}}\right)^2}}{H_1\left[\ln\left(\frac{H_1}{z_0}\right) - 1\right] + z_0}. \tag{25}$$

To include the directional information, we assume that the background velocity profile used to initialize the 3LM, characterized in principle by an angle profile $\Phi(z)$, is rotated by a perturbation angle that only depends on the horizontal location, which we evaluate as

$$\phi'(x,y) = \arctan\left(\frac{V_1 + v_1^{\text{bk}}}{U_1 + u_1^{\text{bk}}}\right) - \arctan\left(\frac{V_1}{U_1}\right) \tag{26}$$

so that the overall flow angle at a given location is $\phi(\mathbf{x}) = \Phi(z) + \phi'(x,y)$. The background velocity profile inside the wind farm layer can finally be expressed as

$$\mathbf{U_b}(\mathbf{x}) = \frac{u^*(x,y)}{\kappa}\left(\ln\left(\frac{z}{z_0}\right)\cos\left(\phi(\mathbf{x})\right)\mathbf{i} + \ln\left(\frac{z}{z_0}\right)\sin\left(\phi(\mathbf{x})\right)\mathbf{j}\right). \tag{27}$$

Note that this procedure can also be used in those cases where the temperature profile is stable or unstable below $H$ by including expressions for $\Psi$ and matching the integration numerically. Moreover, we note that such reconstruction from depth-averaged to actual velocity only matches the vertical average of Equation (23) in the wind farm layer. Adding the extra constraint of matching also the upper layer would result in an over-determined system, which could still be solved in a least-squares sense. As this is out of the scope of the present study, we use Equations (25) to (27) to reconstruct $\mathbf{U_b}(\mathbf{x})$ from $\mathbf{u}_1^{\text{bk}}$.

### 2.2.3 New Wake Superposition

The framework described in Section 2.2 requires a wake superposition method that accommodates the heterogeneous background velocity field $\mathbf{U_b}(\mathbf{x})$. Brogna et al. (2020) developed a variation of Equation (16) that accounts for a variable background wind speed and direction, while Farrell et al. (2021) combined wake deficits using Equation (15) with $U_\infty = u_k =$





$\|\mathbf{U_b}(\mathbf{x})\|$. Lanzilao and Meyers (2022a) also developed new superposition techniques for combining wake deficits with a background flow with variable magnitude or both variable magnitude and direction. Although these methods are all compatible with the MSC framework, we introduce a very simple superposition strategy, which has the advantage of being equivalent to Equation (15) or Equation (16) when the background flow is uniform in both speed and direction. This ensures that wake

deficits are combined consistently with previous literature, avoiding the uncertainty on whether or not the wake model must be re-tuned.

First, we assume that the background flow angle varies on a much larger spatial scale than the turbine spacing. As a result, for a given turbine $k$, the wake is assumed to be aligned with the local background flow at the rotor, which is rotated accordingly. As a consequence, $W = W(\mathbf{x}|_k)$, where $|_k$ means that $\mathbf{x}$ is expressed based on the local wake coordinate system of turbine $k$

(see Section 2.1.2). In addition, we set $U_\infty = \|\mathbf{U_b}(\mathbf{x})\|$. With the above variations, Equations (15) and (16) become

$$U_w(\mathbf{x}) = \left\|\mathbf{U_b}(\mathbf{x})\right\| - \sum_{k=1}^{Nt} u_k W(\mathbf{x}|_k) \tag{28}$$

$$U_w(\mathbf{x}) = \left\|\mathbf{U_b}(\mathbf{x})\right\| - \sqrt{\sum_{k=1}^{Nt} \left(u_k W_k(\mathbf{x}|_k)\right)^2}, \tag{29}$$

where $U_w(\mathbf{x})$ now contains the combined effect of turbine wakes and large-scale pressure gradients. Equations (28) and (29) are different to the formulation of Farrell et al. (2021) in that $u_k$ is equal to the average of $U_w(\mathbf{x})$ among quadrature points of

turbine $k$, rather then the free stream velocity at location $\mathbf{x}$. Moreover, it is worth noticing that $U_w(\mathbf{x})$ is, in reality, a velocity magnitude, as the directional information contained in $\mathbf{U_b}(\mathbf{x})$ only enters in the above equations by varying the direction of the wake shed by a given wind turbine. In the present study, we follow the combination of wake and turbulence intensity models used by Niayifar and Porté-Agel (2016) and Allaerts and Meyers (2019), who both used Equation (15). In the context of the MSC model, to remain consistent with such combination, we use Equation (28) to combine wake deficits with the varying

background wind $\mathbf{U_b}(\mathbf{x})$. Finally, we include ground effects by mirroring each wind turbine with respect to the ground plane (Nygaard et al., 2022).

### 2.2.4 Local Blockage Model

Different local blockage models have been developed by the research community (see Branlard et al. (2020) for a review). Any of these models is compatible with the MSC model framework as long as they satisfy the superposition and separation-of-scales

principles. In the present study, we choose to model local blockage using the vortex cylinder model (Branlard and Gaunaa, 2014; Branlard and Meyer Forsting, 2020). To this end, referring to the local wake coordinate system of turbine $k$, the axial perturbation velocity produced by turbine $k$ at given point is given by

$$u_k^{vc}(r, x') = \frac{\gamma_t}{2\pi} \left[ \frac{R - r + |R - r|}{2|R - r|} + \frac{x' k(r, x')}{2\pi \sqrt{rR}} \left( K\left(k^2(r, x')\right) + \frac{R - r}{R + r} \Pi\left(k^2(r, 0), k^2(r, x')\right) \right) \right], \tag{30}$$

where $x' = x - x_k$ is the axial distance from the rotor and $r = \sqrt{(y - y_k)^2 + (z - z_k)^2}$ is the radial distance from the wake

axis (with $x_k$, $y_k$ and $z_k$ the coordinates of the rotor center of turbine $k$). $R$ is the turbine radius, $K$, and $\Pi$ are the complete





elliptic integrals of the first and third kind respectively, with elliptic parameter

$$k(r, x') = \frac{4rR}{(R+r)^2 + x'^2}.$$ (31)

The momentum theory calibration relation (Branlard and Meyer Forsting, 2020) is used to express $\gamma_t$ as

$$\gamma_t = -\|\mathbf{U_b}(\mathbf{x}_k)\| \left(1 - \sqrt{1 - C_T}\right).$$ (32)

The total velocity at a point, containing wake and local induction effects, is then

$$U(\mathbf{x}) = U_w(\mathbf{x}) + \sum_{k=1}^{N_t} u_k^{vc}(r, x').$$ (33)

Ground effects are included by mirroring the vortex cylinder with respect to the ground plane. Local induction at a given point is then computed using Equation (33), i.e. by summing each turbine's individual effect, including images. Regarding such operation it is important to notice that, at each coupling iteration, the final velocity $U(\mathbf{x})$, containing meso- and micro-

scale effects and averaged among each turbine quadrature points, is used to update both turbine thrust, thrust coefficient $C_T$, and power coefficient $C_P$. These operations require that $U(\mathbf{x})$ is the free stream wind speed for the wind turbine of interest, which becomes a concept somewhat ambiguous in waked conditions. In this paper, we follow the approach of Branlard and Meyer Forsting (2020), where thrust, $C_T$ and $C_P$ are evaluated using a free stream velocity that contains induction effects from all cylinders except the ones associated to the current and mirrored turbines. We choose this method as it is consistent

with what has been commonly done so far in the context of engineering models. We also tried another approach exploiting 1D momentum theory (in line with the methodology followed in most LES studies), but the results are similar (not shown here). In this approach, self-induction is retained in the model, which returns at this point the averaged disk velocity $U_{disk}$ for each turbine. Then, the hypothetical free stream wind speed is calculated as $U_\infty = U_{disk}/(1-a)$, where $a = (1 - \sqrt{1 - C_T})/2$ is the axial induction factor. This free stream wind speed $U_\infty$ is then used to update $C_T$ and $C_P$ coefficients from the turbine

manufacturer curves. Finally, thrust and power are calculated using the disk-averaged velocity and the disk-based thrust and power coefficients proposed by Meyers and Meneveau (2010).

### 2.2.5 Solution Procedure

This section details the overall MSC model solution procedure, which is also sketched in Figure 4. The evaluation of the background state is described in Section 2.3 as it is not strictly part of the model and can be attained in different ways. As an

initialization step, the wake model is first run using a uniform inflow velocity (optionally adding local induction effects) and initial thrust and $C_T$ are calculated for each turbine. At this point, the coupling loop is entered, and all models are solved in sequence at each iteration. Specifically, turbine thrust is filtered on the 3LM grid using Equation (18), then Equations (1), (2) and (5) are numerically solved, yielding perturbation fields $\mathbf{u_1}$, $\mathbf{u_2}$, $\eta$ and $p$. In Fourier space, these equations represent a linear system with a coefficient matrix that is non-symmetric and block-diagonal (see Appendix C for details). Equation (21) and

Equation (22) are then used to solve for $\mathbf{u}^{bk}$, and Equations (25) to (27) are used to reconstruct the background velocity field





$\mathbf{U}_b(\mathbf{x})$ at turbine quadrature points from the 3LM grid by means of bi-linear interpolation formulas. Successively, the wake model is run to yield the average of $U_w(\mathbf{x})$ at the quadrature points. In particular, combining Equations (17) and (28) yields

$$\frac{1}{N_q}\sum_{q=1}^{N_q} U_w(\mathbf{x}_{i,q}) + \sum_{\substack{j\\j\neq i}}^{N_t}\frac{1}{N_q}\sum_{q=1}^{N_q} U_w(\mathbf{x}_{j,q})W(\mathbf{x}_{i,q}|j) = \frac{1}{N_q}\sum_{q=1}^{N_q}\big\|\mathbf{U}_b(\mathbf{x}_{i,q}|j)\big\| \tag{34}$$

which produces the linear system $Au^W = b$ where

$$u_i^W = \frac{1}{N_q}\sum_{q=1}^{N_q} U_w(\mathbf{x}_{i,q}) \tag{35}$$


$$A_{ij} = \begin{cases} 1 & i = j \\ \sum_{\substack{j\\j\neq i}}^{N_t}\frac{1}{N_q}\sum_{q=1}^{N_q} W(\mathbf{x}_{i,q}|j) & i \neq j \end{cases} \tag{36}$$

$$b_i = \frac{1}{N_q}\sum_{q=1}^{N_q}\big\|\mathbf{U}_b(\mathbf{x}_{i,q}|j)\big\|. \tag{37}$$

In Equations (34) to (37), the notation $\mathbf{x}_{i,q}|_j$ denotes quadrature point $q$ of turbine $i$, seen in the local wind coordinate system of turbine $j$.

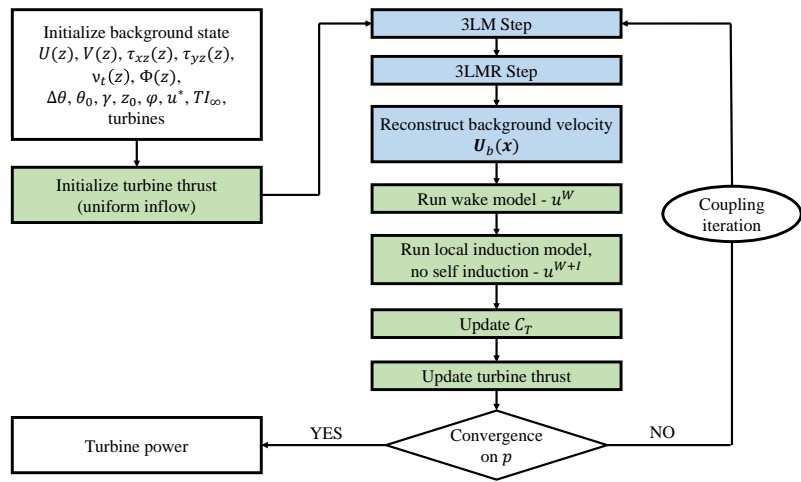

**Figure 4.** Sketch of the MSC model solution procedure. We adopted the same colors of Figure 3 to denote the micro-scale and meso-scale levels of the model.





At this point, local induction effects are added to $u_i^W$ without considering turbine self-induction. Using Equation (30), and averaging on quadrature points yields

$$u_i^{W+I} = u_i^W + \sum_{\substack{j \\ j \neq i}}^{N_t} \frac{1}{N_q} \sum_{q=1}^{N_q} u_j^{vc}(r_{i,q}, x_{i,q}|_j), \tag{38}$$

where $r_{i,q}$ and $x_{i,q}|_k$ are the radial and axial distances between the quadrature point $q$ of turbine $i$ and the rotor axis and center of turbine $j$ respectively. Finally, the $C_T$ coefficient of each turbine is updated using $u_i^{W+I}$ and the input $C_T$ curves, while

turbine thrust is evaluated with Equation (20), where we set $u_k = u_k^{W+I}$. The coupling iteration is then repeated until the L2 norm of the relative pressure residual is lower than a specified tolerance. For instance, we noticed that around 4 or 5 iterations are needed to satisfy a specified tolerance of $10^{-4}$.

### 2.3 Background State

The MSC model requires information about the background ABL state, as well as the wind farm layout and turbine data.

Regarding the first set of information, the required input parameters are reported in Table 1.

| input parameter | explanation |
| --- | --- |
| $\rho, g$ | flow density [kg/m$^3$], gravity [m/s$^2$] |
| $H, H_1, H_2$ | inversion, wind farm and upper layer heights respectively [m] |
| $\Delta\theta$ | inversion strength [K] |
| $\theta_0$ | reference potential temperature [K] |
| $\gamma$ | lapse rate [K/m] |
| $\phi$ | latitude [deg] |
| $z_0$ | equivalent roughness length [m] |
| $u^*$ | background friction velocity [m/s] |
| $\nu_{t,1}, \nu_{t,2}$ | average eddy viscosity in the wind farm and upper layer [m$^2$/s] |
| $\mathbf{U_1}, \mathbf{U_2}, \mathbf{U_3}$ | average background velocity vector in each layer [m/s] |
| $\Phi(z)$ | background wind angle profile [deg] |
| $\left\|\tau\|_{z=0}\right\|, \left\|\tau\|_{z=H_1}\right\|$ | background shear stress magnitude at the wall and at $H_1$ [m$^2$/s$^2$] |
| $TI_\infty$ | free stream turbulence intensity in the ABL [$-$] |

**Table 1.** Input parameters for the MSC model.

In the context of the present study, we chose $H_1$ as twice the average turbine hub-height, while $H_2 = H - H_1$. Reference potential temperature, inversion jump and lapse rate can be prescribed based on observations, while latitude and roughness length can be estimated depending on the specific wind farm site. Average background velocities, wind angle and shear stress magnitudes at the ground and at $z = H_1$ require knowledge about the veered velocity and shear stress magnitude profiles.





These can again be evaluated from observations (Allaerts et al., 2018), from analytical models (see for example Nieuwstadt, 1983) or from numerical simulations. The same holds for background friction velocity and ambient turbulence intensity at the hub-height. The depth-averaged eddy viscosity profiles can be calculated according to Nieuwstadt (1983) as

$$
\nu_t(z) = \begin{cases} \kappa u^* z \left(1 - \frac{z}{H}\right)^2 & z < H \\ 0 & z >= H \end{cases}.
\tag{39}
$$

In the present study, as our aim is to validate the MSC model against wind farm LES, we evaluate the background state using results from our conventionally-neutral boundary layer (CNBL) precursor simulations. This avoids any mismatch in the initial background state, making sure that any discrepancy in turbine performance is due to the model only.

## 3 High-Fidelity Validation Data

The present section describes methodology and results of the LES simulations used to validate the MSC framework. First, the adopted numerical set-up is described in Section 3.1. Then, Section 3.2 reports the choice of input parameters used for the CNBL simulations, characterizing the resulting background atmospheric state. The latter is used both to define the background ABL states required by the 3LM and MSC models, as well as to prescribe the inflow condition for the wind farm analyses discussed in Section 3.3.

### 3.1 LES Methodology

For the LES simulations presented in this paper, we use the open-source finite volume code TOSCA (Toolbox fOr Stratified Convective Atmospheres) developed at the University of British Columbia. Details about the code, governing equations and LES methodology are thoroughly explained in Stipa et al. (2023). For the sake of brevity, here we only report the adopted setup for the wind farm simulations.

To avoid gravity wave reflections, we use both periodic boundary conditions and a fringe region located at the domain inlet. This also provides a suitable turbulent inflow, eliminating the wind farm wake re-advected at the inlet by periodic boundaries. At the upper boundary, we use a Rayleigh damping layer with a thickness of 12 km, i.e. slightly more than one expected vertical wavelength (this parameter can be calculated as $\lambda_z = 2\pi U_g/N$, where $N$ is the Brunt-Väisälä frequency and $U_g$ is the geostrophic wind). Lateral boundaries are periodic, implying that gravity waves induced by the wind farm will interact with their periodic images. This requires the domain to be sufficiently large for these interactions to happen far from and downstream of the wind turbines. Moreover, we use the advection damping technique developed by Lanzilao and Meyers (2022b) to ensure that interactions between fringe-generated and physical waves are not advected downstream but remain trapped inside the advection damping region. After a reflectivity study that employed a computationally cheap canopy model (not shown here), we found that a Rayleigh damping coefficient of $\nu_{RDL} = 0.05~s^{-1}$ and a fringe damping coefficient of $\nu_{FR} = 0.03~s^{-1}$ yielded minimal gravity wave reflection. We used the same damping functions as Lanzilao and Meyers (2022b), and in Table 2 we report their parameters for our wind farm simulations.





| $z_s$ [km] | $z_e$ [km] | $\Delta z$ [m] | $N$ [-] | $f$ [-] |
|---|---|---|---|---|
| 0 | 0.4 | 10 | 40 | 1 |
| 0.4 | 0.5 | 10-4.85 | 14 | 0.94591 |
| 0.5 | 0.6 | 4.59-10 | 15 | 1.05125 |
| 0.6 | 1 | 10 | 40 | 1 |
| 1 | 3 | 10-100 | 51 | 1.04698 |
| 3 | 17 | 100 | 140 | 1 |
| 17 | 28 | 100-500 | 44 | 1.03818 |

(a) Vertical discretization parameters.

| $x_s$ [km] | $x_e$ [km] | $\Delta x$ [m] | $N$ [-] | $f$ [-] | $y_s$ [km] | $y_e$ [km] | $\Delta y$ [m] | $N$ [-] | $f$ [-] |
|---|---|---|---|---|---|---|---|---|---|
| -20 | -15.005 | 15 | 333 | 1 | -9 | -1.5 | 20 | 375 | 1 |
| -15.005 | -13 | 15-30 | 94 | 1.00748 | -1.5 | -0.5 | 20-12.5 | 62 | 0.99269 |
| -13 | 18.02 | 30 | 1035 | 1 | -0.5 | 3.5 | 12.5 | 320 | 1 |
| 18.02 | 19.97 | 30-15 | 90 | 0.9923 | 3.5 | 4.5 | 12.5-20 | 62 | 1.00805 |
| 19.97 | 20 | 15 | 2 | 1 | 4.5 | 12 | 20 | 375 | 1 |

(b) Streamwise (left) and spanwise (right) discretization parameters.

**Table 3.** Mesh information for the wind farm cases.

| $x_s$ [km] | $x_e$ [km] | $\Delta_s$ [km] | $\Delta_e$ [km] |
|---|---|---|---|
| $-20$ | $-15$ | 1 | 1 |

(a) Fringe region parameters.

| $x_s$ [km] | $x_e$ [km] | $\Delta_s$ [km] | $\Delta_e$ [km] |
|---|---|---|---|
| $-18$ | $-11$ | 1 | 1 |

(b) Advection damping region parameters.

**Table 2.** Fringe and advection damping region information.

The size of the successor domain is 40 km × 21 km × 28 km in the streamwise, spanwise and vertical direction respectively, discretized with 1554 × 1194 × 345 cells. All directions are graded to reach a mesh resolution of 30 m × 12.5 m × 10 m around the wind farm. The concurrent precursor mesh coincides with the portion of the successor domain located inside the fringe region. As a consequence, it extends for 5 km × 21 km × 28 km. The mesh resolution in the streamwise direction is 15 m, while in the spanwise and vertical directions, it is the same as the successor. Detailed mesh information is reported in
Table 3.

     To obtain a fully-developed turbulence state in the concurrent precursor, cutting down computational cost at the same time, we use the hybrid off-line/concurrent precursor technique described in Stipa et al. (2023). Specifically, a separate off-line precursor is first conducted on a domain smaller than the fringe region for $1.2 \cdot 10^5$ s, and inflow slices are saved at each time step during the last 20k s of simulations. This first phase defines the background inflow conditions, and it is described



with more detail in Section 3.2. Inflow slices are then periodized along the spanwise direction and mapped at the concurrent precursor inlet, as it uses inflow-outflow boundary conditions in this initial phase. After one concurrent precursor flow turnover time, inlet and outlet boundaries are switched to periodic, and the solution is then progressed for 30k s. Note that successor and concurrent precursor always proceed in sync with each other, and the former always employs periodic boundary conditions at the streamwise boundaries. After one successor flow turnover time ($\approx$ 5k s) flow statistics are gathered for 25k s.

The wind farm has a rectangular planform, with 20 rows and 5 columns. The first row is located at $x = 0$, and extends from 300 m to 2700 m in the spanwise direction. This determines a lateral spacing of 600 m (4.76 D), while streamwise spacing is set to 630 m (5 D). Wind turbines correspond to the NREL 5-MW reference turbine, and are equipped with angular velocity and pitch controllers described in Jonkman et al. (2009). A very simple yaw controller is also added, which rotates wind turbines independently using a uniform speed of 0.5 deg/s when flow misalignment exceeds 1 deg. Flow angle is calculated by filtering

the wind velocity at a sampling point located 1D upstream of the rotor center, using a time constant of 600 s. Turbines are modeled using the actuator disk model (ADM) described in Stipa et al. (2023), while tower and nacelle are not included. The ADM force projection width is set to 18.75 m.

In both the concurrent precursor and successor simulations, velocity is controlled using a constant source term, obtained by averaging the off-line precursor sources for the last 20k s. Temperature controller is retained in the concurrent precursor

simulation, but it is switched off in the successor to allow the inversion height to be freely perturbed by the wind farm.

## 3.2 CNBL Precursors

In order to highlight the differences between subcritical and supercritical regimes, at the same time isolating the impact that atmospheric stability has on wind farm efficiency, we simulate two CNBL cases that only differ in the capping inversion strength, but that have the same wind profile and free atmospheric lapse rate. The choice of input parameters, summarized in

Table 4 for both simulations, is based on the sensitivity study performed by Allaerts and Meyers (2019). We set the inversion strength of the subcritical case (N1) to 7.312 K, corresponding to $F_r \approx 0.9$, while we chose a value of 4.895 K, corresponding to $F_r \approx 1.1$, for the supercritical case (N2). The Coriolis parameter $f_c$ corresponds to a latitude of 41.33 deg.

| $u_{ref}$ [m/s] | $h_{ref}$ [m] | $\theta_0$ [K] | $\Delta h$ [m] | $\gamma$ [K/km] | $H$ [m] | $f_c$ [1/s] | $z_0$ [m] |
|---|---|---|---|---|---|---|---|
| 9.0 | 90 | 300 | 100 | 1 | 500 | $9.6057 \cdot 10^{-5}$ | 0.05 |

**Table 4.** ABL parameters used for the finite wind farm simulation presented in this section

The Rampanelli and Zardi (2004) model is used to initialize the potential temperature profile, where $H$ is taken as the center of the capping inversion layer. The two CNBL precursors are advanced in time for $10^5$ s, then data are averaged for $2 \cdot 10^4$ s.

The domain size of the two CNBL simulations is of 6 km $\times$ 3 km $\times$ 1 km in the streamwise, spanwise, and vertical directions respectively. The mesh has a horizontal resolution of 15 m, while in the vertical direction it is graded equally as the concurrent precursor and successor simulations. A driving pressure controller with geostrophic damping is used to fix the average velocity



at $h_{\text{ref}}$, chosen as the hub height, while a potential temperature controller is used to fix the average potential temperature profile throughout the simulation (both controllers and geostrophic damping use the same settings reported in Stipa et al., 2023).

Figure 5 shows the resulting profiles of velocity magnitude, wind angle, potential temperature and shear stress from simulations N1 and N2. In Table 5, quantitative data extracted from the two simulations are also reported. The capping inversion center $H$, ground temperature $\theta_0$, inversion strength $\Delta\theta$ and inversion width $\Delta h$ are calculated by fitting the Rampanelli and Zardi (2004) model in a least-squares sense. It is clear both from Table 5 and Figure 5 that cases N1 and N2 have almost identical wind and shear stress profiles, while the background temperature stratification differs in the inversion strength, as explained earlier. Moreover, the pressure and potential temperature controllers allow us to obtain both an average hub-height wind speed and a ground potential temperature that match the input parameters listed in Table 4. According to the obtained CNBL profiles, it is clear that wake models alone (optionally combined with a local induction model) would predict very similar power production for each individual turbine in the two cases. In reality, as will be shown later, wind farm interaction with the thermally stratified boundary layer leads to very different trends in power production within the farm.

| | $u_{ref}$ [m/s] | $G$ [m/s] | $\theta_0$ [K] | $\Delta\theta$ [K] | $\Delta h$ [m] | $H$ [m] | $u^*$ [m/s] | $q_{min}/10^{-4}$ [Km/s] | $\phi_G$ [deg] |
|---|---|---|---|---|---|---|---|---|---|
| N1 | 8.98 | 10.60 | 300 | 7.312 | 98 | 500 | 0.430 | -1.04 | -24.7 |
| N2 | 8.98 | 10.68 | 300 | 4.895 | 95 | 500 | 0.431 | -1.04 | -23.5 |

**Table 5.** ABL parameters obtained by fitting the Rampanelli and Zardi (2004) model for the CNBL cases presented in this section, together with resulting friction velocity, minimum heat flux and geostrophic wind angle.

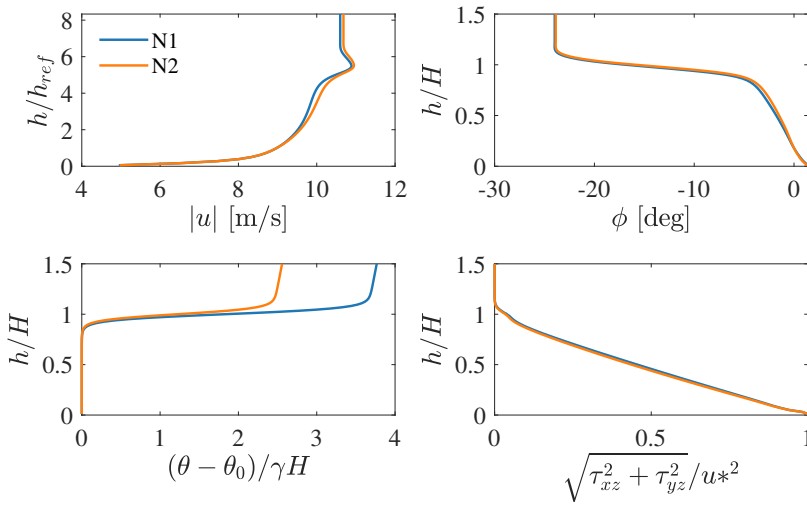

**Figure 5.** Velocity magnitude, wind direction, potential temperature, and shear stress profiles from precursor simulations.



## 3.3 Wind Farm Successors

In the current section, we briefly present results from our wind farm large-eddy simulations. Figure 6 shows instantaneous fields of velocity magnitude at the hub height, for both the subcritical and supercritical cases, at the end of each simulation. High wake meandering can be observed in both cases, together with the gradual formation of a wind farm wake when moving downstream of the first row. However, from this figure it is not possible to appreciate the differences existing between the two simulations at the large scale. Therefore, it is instructive to look at the perturbation pressure field, which is evaluated by computing the difference between the pressure experienced in the successor domain at a given height and horizontal location, and the horizontally-averaged pressure from the concurrent precursor, evaluated at the same height. This quantity is shown in Figure 7, for the physical portion of the successor domain, i.e. excluding the fringe and advection damping regions. Substantial differences can be noticed between the two cases. In particular, the subcritical case (N1) shows a broader region of pressure increase with respect to the supercritical case (N2), with its maximum located around the first-row turbines. While in both cases such positive pressure anomaly lasts for the first few wind farm rows, it extends for a longer distance upstream of the turbines in case N1. Additionally, the local and individual pressure increase in front of each wind turbine can also be observed. At the end of the wind farm, both cases show a negative pressure anomaly, less substantial than the positive one in front, and again more pronounced in the subcritical case.

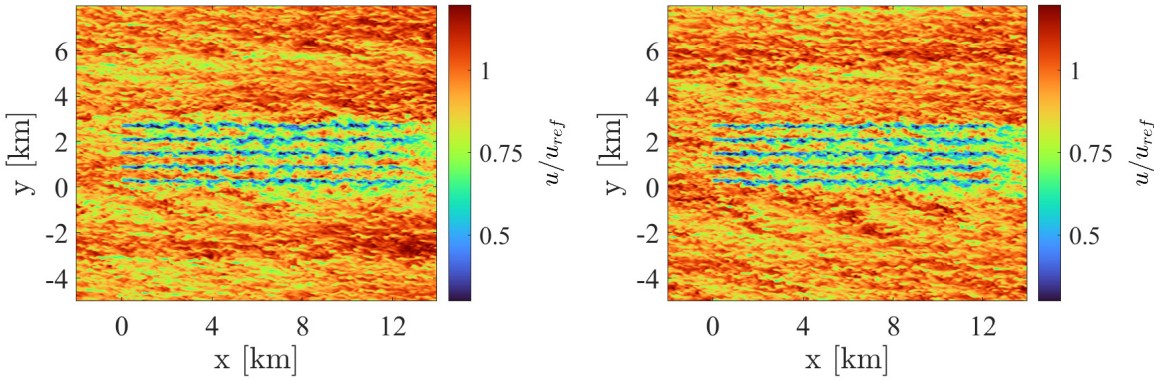

**Figure 6.** Instantaneous velocity magnitude at the hub-height, sampled at the end of wind farm cases N1 (left) and N2 (right).

Inside the wind farm, a generally favorable pressure gradient is observed. The formation of these pressure anomalies can be explained by looking at the wind farm as an obstacle to the approaching flow. First, streamlines are displaced upwards and downwards at the wind farm entrance and exit, respectively. In addition, the boundary layer has to displace upwards after the first turbine rows in order to compensate for the mass flux deficit produced by wind turbine wakes. This results in a displacement of the inversion layer near the wind farm start that is larger than what is observed at the wind farm exit. Then, as the flow is vertically stratified starting from the inversion height, an upward perturbation of the flow particles corresponds to a positive pressure anomaly felt below, as a taller column of dense air is locally overtopping the ground. Furthermore,




these vertical perturbations trigger interfacial and internal waves in the inversion layer and atmosphere aloft, respectively. As reported by Lin (2007), the interfacial wave amplitude depends on the $F_r$ number. In subcritical conditions, these waves can propagate upstream, determining an increase of layer depth before the obstacle, thus augmenting the total energy of the flow. This cannot happen in supercritical conditions, as the interfacial wave speed is smaller than the advection speed. In such condition, upstream propagation is produced by internal waves only, and smaller anomalies of the inversion layer in terms of both the amplitude and spatial scale are observed.

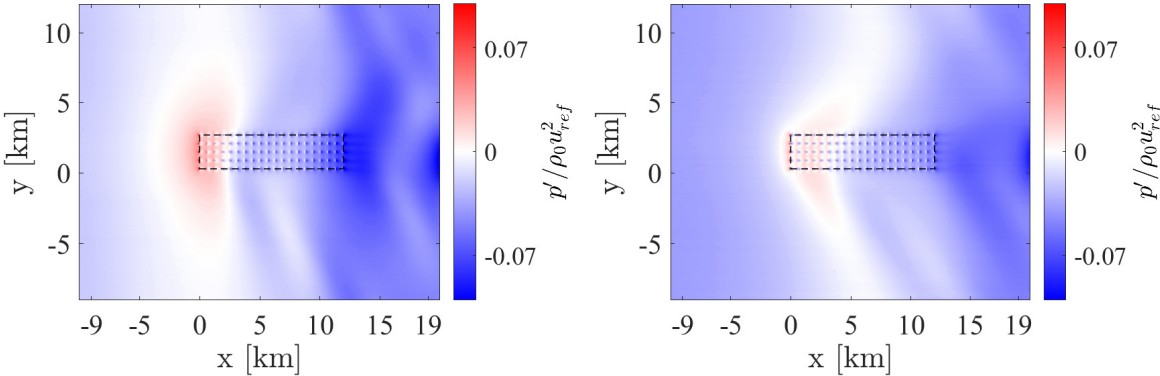

**Figure 7.** Perturbation pressure at the hub-height for wind farm cases N1 (left) and N2 (right).

The positive correlation that exists between the boundary layer displacement and pressure perturbations are especially clear in Figure 8. Here, perturbation pressure is shown in combination with the streamline displacement at the inversion height (magnified 5 times), which is a good estimation of the position of the inversion center locally (Allaerts and Meyers, 2017).

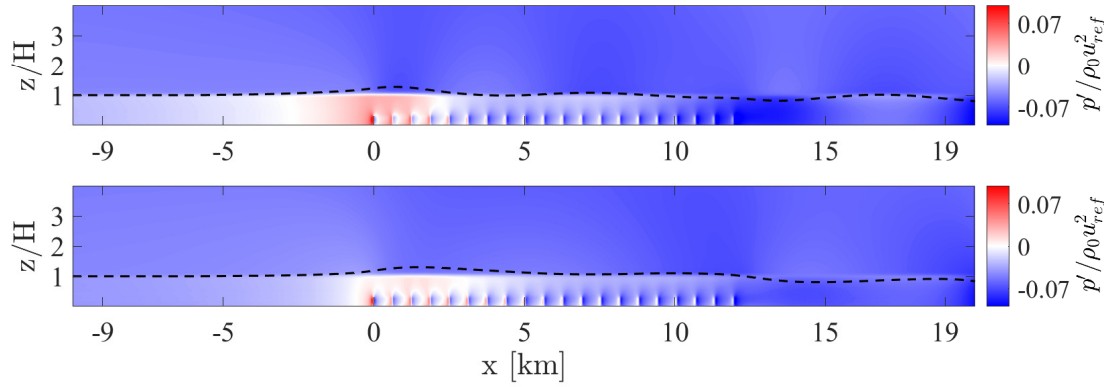

**Figure 8.** Perturbation pressure below the inversion layer for wind farm cases N1 (top) and N2 (bottom). The dashed line indicates the displacement of the streamline located at $z = H$ at the inlet, magnified 5 times.





Moreover, it is clear that the pressure field around the wind farm is more complex than the mere superposition of individual turbine effects, but is regulated by the interaction between the wind farm and the stratified boundary layer. It is reasonable to assume that such large-scale pressure disturbances around the farm also perturb the wind speed. Though such an effect is too small to be visualized in Figure 6, it can affect wind farm power in different ways. First, the adverse pressure gradient upstream

of the first row decreases the approaching wind and, depending on thermal stratification, different levels of global blockage will be experienced. Inside the wind farm, despite oscillations in the subcritical case, the pressure gradient is always favorable, and counteracts the wind speed reduction. In the wind farm wake, especially for the subcritical case, the gravity waves clearly induce pressure oscillations, which may result in an apparently intermittent or non-monotonic wake recovery.

We emphasize that the two simulations share an almost identical wind profile, and only differ in the background thermal

stratification. The very different observed behavior, only due to differences in the potential temperature profile, is currently not captured by engineering models. In fact, even if modeling of the wind farm blockage had been attempted e.g. by superimposing individual induction effects via, e.g., the vortex cylinder (Branlard and Gaunaa, 2014) or the Rankine half body (Gribben and Hawkes, 2019) approach, the models would erroneously produce identical results for wind farm cases N1 and N2. These LES results highlight the importance of accounting for atmospheric stability in engineering models.

**4  MSC Model Results and Validation**

In this section, results from our wind farm LES simulations are compared against the original 3LM and the newly developed MSC model. For reference, we also show results obtained using the wake model alone (current industry standard), and the combination of the latter with the vortex cylinder model, to include local blockage effects. The validation of the model is performed by comparing its predictions against LES, focusing on both the flow variables, in Section 4.1, and on turbine

quantities in Section 4.2.

In Table 6, we report the input parameters used to define the two background states for the meso-scale sub-model. The wind angle profile coincides with the one shown in Figure 5. For the background height-averaged velocities in each layer, or to set the free stream velocity for the wake model with and without local induction, we use the inflow velocity profile at the end of the advection damping region (the actual start of the meaningful portion of the successor domain). We observed that this velocity

profile differs slightly (3.5% at hub height) from the profile in the concurrent precursor simulation (this was also observed by Lanzilao and Meyers, personal communication). It is not clear why the fringe region induces such a modification of the input profile, and this is a topic for further investigation.





| input parameter | N1 | N2 | |
|---|---|---|---|
| $g$ | 9.81 | | [m/s$^2$] |
| $\rho$ | 1.225 | | [kg/m$^3$] |
| $H, H_1, H_2$ | 500, 180, 320 | | [m] |
| $\Delta\theta$ | 7.312 | 4.895 | [K] |
| $\theta_0$ | 300 | | [K] |
| $\gamma$ | 1 | | [K/km] |
| $\phi$ | 41.33 | | [deg] |
| $z_0$ | 0.05 | | [m] |
| $u^*$ | 0.43 | | [m/s] |
| $\nu_{t,1}, \nu_{t,2}$ | 9.37, 6.19 | | [m$^2$/s] |
| $(U_1, U_2, U_3)$ | (8.31, 10.07, 9.77) | (8.41, 10.32, 10.16) | [m/s] |
| $(V_1, V_2, V_3)$ | (−0.05, −0.78, −4.49) | (0.09, −0.2, −4.41) | [m/s] |
| $\left\|\tau\|_{z=0}\right\|, \left\|\tau\|_{z=H_1}\right\|$ | 0.19, 0.11 | | [m$^2$/s$^2$] |
| $TI_\infty$ | 0.09 | | [−] |

**Table 6.** 3LM input parameters for cases N1 and N2.

As described in Section 2.2.3, wind turbines are mirrored in the MSC model, while in all other models we do not perform such operation. In fact, we noticed that the trend of row-averaged thrust and power distributions is better captured in these cases without mirroring effects. Conversely, the addition of turbine images would produce increased wake effects, leading to thrust and power distributions which are monotonically decreasing with downstream distance, after the first few rows. This would result in a thrust and power over prediction at the first rows, while these quantities would be strongly under predicted towards the wind farm end. We observed that the increased wake effects produced by rotor images tend to balance the overestimation given by the absence of blockage at the first rows, reducing the error in overall wind farm power. This is a lucky situation, where two model issues counter-balance each other to reduce the global error while completely misrepresenting the actual shape of thrust and power distributions. Choosing not to include mirroring effects is in agreement with many previous literature studies, and highlights how, despite capturing the overall trend, power and thrust distributions are consistently shifted towards higher values because of a first-row bias that arises when global blockage is neglected. Regarding the MSC model, the favorable pressure gradient produced by gravity waves between the first and last wind farm rows causes the background velocity to accelerate slightly within the wind farm, depending on the specific stability conditions. For this reason, increased wake effects produced by turbine images are not an issue in this case, and ground effects can be modeled without affecting the shape of power and thrust distribution.

Regarding the 3LM, we use a numerical grid of $400 \times 203$ km in the streamwise and spanwise direction respectively, discretized with a uniform $500 \times 500$ m grid. As the NREL 5-MW thrust curve is not available in official literature, we ran





several large-eddy simulations of isolated ADMs with uniform, non-turbulent inflow to compute the thrust and power curves
used in the reduced models for the present analysis (see Appendix B). In both the MSC model and original 3LM runs, we
use 5 coupling iterations, which are sufficient to bring the pressure residual below $10^{-4}$. We noticed that only one iteration
is sufficient to capture the gross wind farm gravity wave interactions (results not shown here), while three iterations provide
thrust and power distributions within $0.5\%$ of the value obtained using 5 iterations.

## 4.1 Flow Field Validation

In this section we compare the flow quantities predicted by the reduced-order models, against the LES results.

Figure 9 shows, for both subcritical and supercritical conditions, the depth-averaged perturbation velocities in the wind farm
layer, calculated after solving the 3LM step — Equations (1), (2) and (5) — and the 3LMR step — Equations (21) and (22). The
background perturbation velocity field $u_1^{\text{bk}}$ is then compared with vertical profiles of $\mathbf{U}_b(\mathbf{x})$, obtained after applying the average
matching procedure — Equations (25) to (27) — for two locations of interest, i.e. the middle of first and last wind farm rows.
First, it can be seen from the reconstructed hub-height velocity, that the region of maximum flow deceleration is located more
upstream in the subcritical than the supercritical case. However, the difference between the two regimes is more evident in the
downstream direction, as the supercritical case is characterized by several distinct V-shaped waves generated by the principal
sources of boundary layer displacement, i.e. the wind farm entrance and exit. Conversely, in the subcritical case, the wind farm
excites resonant lee waves, as the vertical wavenumber $m$ becomes imaginary. These is in agreement with Allaerts and Meyers
(2019) and Devesse et al. (2022). Moreover, in both cases the wind farm wake is deflected towards the negative $y$ direction by
momentum exchange between the wind farm and upper layers. Another interesting aspect is the complex and non-symmetric
wave pattern that arises to either side of the wind farm due to the interaction of two different wave trains departing from the
wind farm entrance and exit respectively. This is a consequence of the wind veer produced by the Coriolis force, which causes
a misalignment between the geostrophic wind and the hub-height velocity. As a consequence, as interfacial and internal waves
are formed around and above $H$, respectively, there exists a spanwise shift between the two wave trains, approximately given
by $L_x \sin(\Phi(H))$, where $L_x$ is the wind farm length in the $x$ direction and $\Phi(H)$ is the flow angle with respect to the $x$-axis at
the inversion center. This phenomenon is very clear in Figure 9c, where the interaction between resonant lee waves is stronger
on the southern rather than the northern side of the wind farm. Moreover, looking at the reconstructed profiles of $\mathbf{U}_b$, it can be
noticed how the subcritical case experiences more blockage than the supercritical case at the wind farm entrance (Figure 9e),
while the background velocity for case N1 is higher than case N2 at the exit of the cluster (Figure 9f).





**Figure 9.** Depth-averaged perturbation velocity in the wind farm layer, obtained after solving Equations (1), (2) and (5) for subcritical (a) and supercritical (b) case. Background perturbation velocity obtained after solving Equations (21) and (22) for subcritical (c) and supercritical (d) case. Velocity profile produced by the average matching procedure at the center of the first (e) and last (f) wind farm rows. Subcritical and supercritical cases are displayed on the same diagram. Dashed lines highlight the hub height $h_{\text{hub}}$, while dotted lines indicate $h_{\text{hub}} \pm D/2$.



Figure 10 displays the comparison of the predicted perturbation pressure and velocity in the wind farm layer between the LES and MSC model, spanwise-averaged from y= 0.3 to y= 2.7 km. Regarding the 3LM step within the MSC (equivalent to the original 3LM formulation), perturbation pressure agrees quite accurately with LES results, while depth averaged perturbation

velocities are overestimated both in the wind farm and upper layer (the latter is not shown).

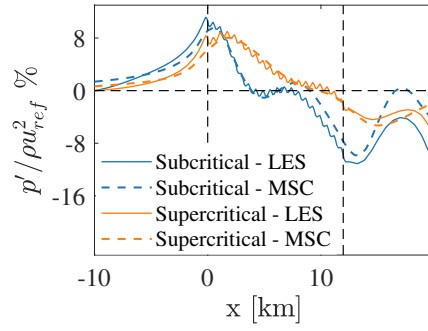
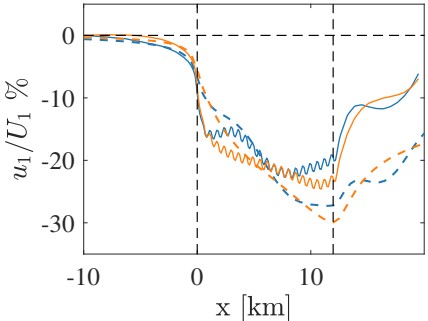

**Figure 10.** Left: perturbation pressure inside the boundary layer given by the 3LM within the MSC and from LES results. LES data are averaged within the upper layer only, to remove the effect of local pressure increase in front of each turbine, not present in the 3LM. Right: depth-average perturbation velocity in the wind farm layer.

As can be noticed, the model predicts a stronger velocity deficit with increasing downstream distance from wind farm start. The same behavior has been observed also by Allaerts and Meyers (2019), who argued that the 3LM underpredicts shear stress, especially at the interface between the wind farm and upper layer, as it neglects the effect of added turbulence intensity produced by the wind farm. While such limitations in capturing velocity deficit and shear stress are topics for further research, we note

that they have almost no influence on the reconstruction step, as only the pressure field is used to compute depth-averaged, large-scale velocity variations. In fact, we observed that pressure-induced wind speed perturbations are small if compared to the ones produced by the wind farm, leading to a negligible magnitude of shear stress terms in the 3LMR step (not shown here). These are mainly retained for numerical reasons, as they improve the conditioning number of the linear system matrix.

Figure 11 compares LES results against the solution obtained at the final step of the MSC model, i.e. the superposition of

$\mathbf{U}_b(\mathbf{x})$ and the analytical wake and induction models. For reference, the velocity resulting from the sole combination of the wake and vortex cylinder models is also shown. First, it can be noticed how the latter approach produces in practice the same results in both subcritical and supercritical conditions, as they only differ slightly in their inflow profiles. Conversely, in the MSC model, large scale pressure perturbations greatly influence the micro scale velocity, making it possible to capture gravity waves flow patterns around and inside the wind farm.

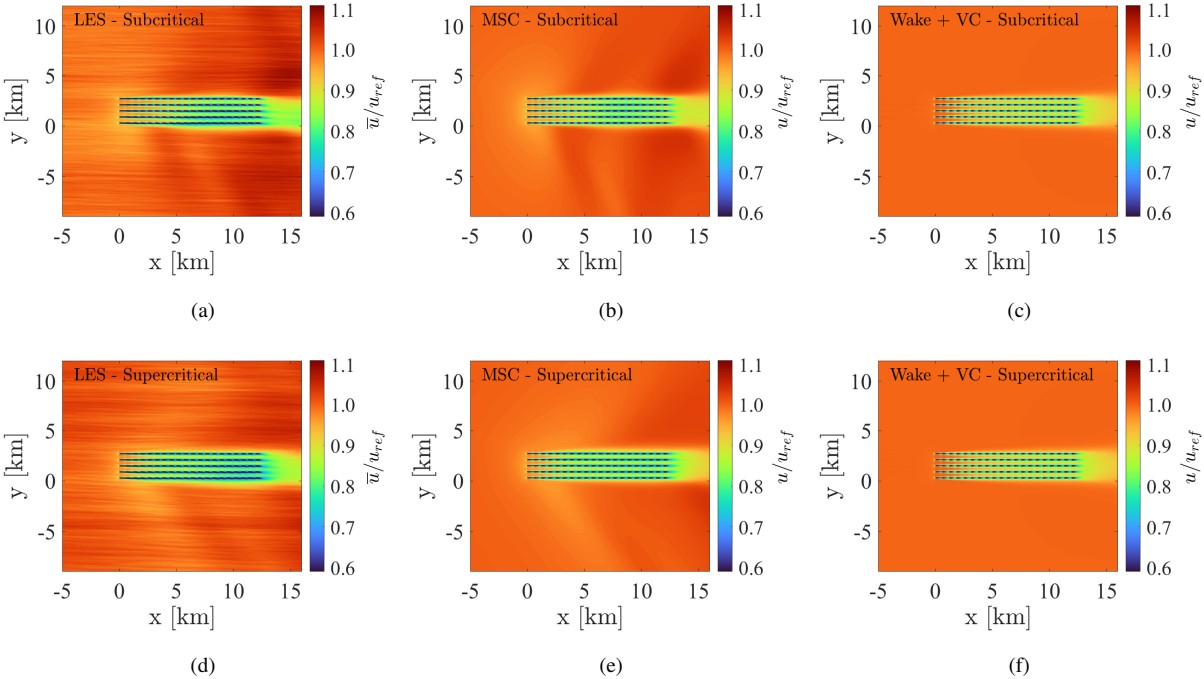

**Figure 11.** Micro-scale flow field obtained from (left) LES, (center) MSC model and (right) wake model with vortex cylinder. Top and bottom diagrams correspond to the subcritical and supercritical case respectively.

In Figure 12, we compare data from Figure 11 against LES results in more detail, for all reduced models. Specifically, we display the row-averaged (spanwise-averaged between the row start and end locations) hub-height velocity for the subcritical case only (data from the supercritical case are not reported, as they point to the same conclusions). Regarding the original 3LM formulation, such analysis was not performed by Allaerts and Meyers (2019), but it is interesting as it allows to visualize what happens inside the wake model, and how blockage effects have been introduced in the original 3LM formulation. In the wakes-only approach, the free stream velocity is constant up to the first wind farm row, as it is not influenced by meso-scale effects. Here it presents a discontinuity due to the presence of the first wake. The same holds for the original 3LM, but the free stream velocity is now reduced due to the 3LM coupling through the sampling point upstream the wind farm. Both models show velocity jumps at the turbine locations due to the discontinuity at the disk given by the wake model. In general, they both predict a higher velocity than LES at the turbine locations, which results in an overestimation of thrust and power, as it will be shown later. Regarding the wake model coupled with the local induction model, it still under-predicts global wind farm blockage, but the velocity distribution is more accurate inside the wind farm. Nevertheless, it is worth remembering that, when using this approach, turbine self-induction is not accounted for in estimating the free stream velocity for a given turbine. Furthermore, local induction effects given by the vortex cylinder model at a given turbine from all remaining turbines are very limited (e.g. blockage is $\approx 1\%$ at 5 diameters upstream, with a $C_T$ of 0.9).





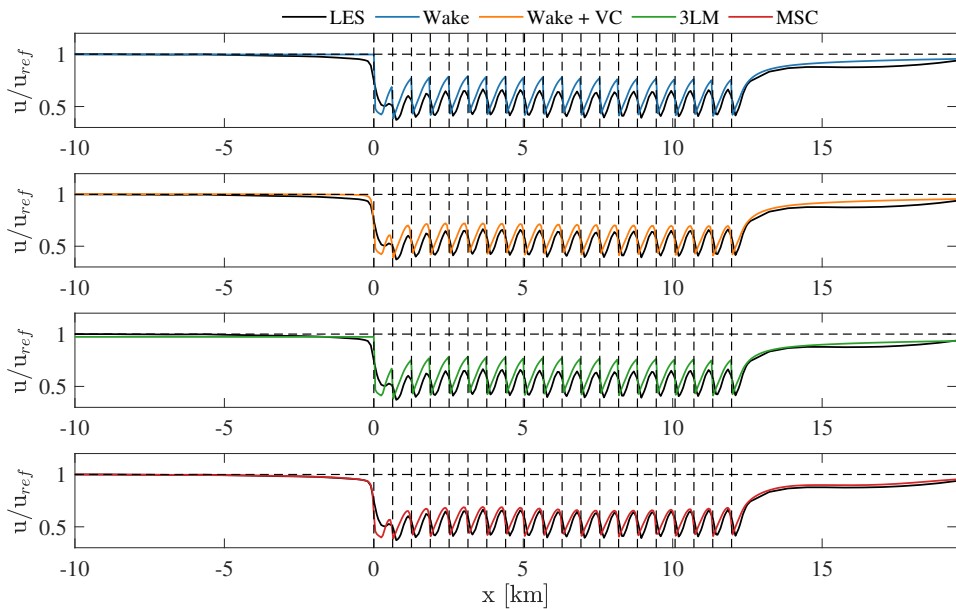

**Figure 12.** Comparison of row-averaged wind between LES and different reduced models. Blue: wake model, orange: wake and local induction model, green: original 3LM, red: MSC model.

For this reason, including local induction effects only produces a negligible improvement in the actual thrust and power estimates if compared with the wakes-only approach. Looking instead at the MSC model, it can be noticed how large and small scale blockage effects are both accurately captured, together with velocity oscillations in the wind farm wake, operated by the large-scale gravity wave-induced pressure gradient. Additionally, the model is also able to satisfactorily capture the velocity distribution inside the wind farm, with the largest error observed just after first row.

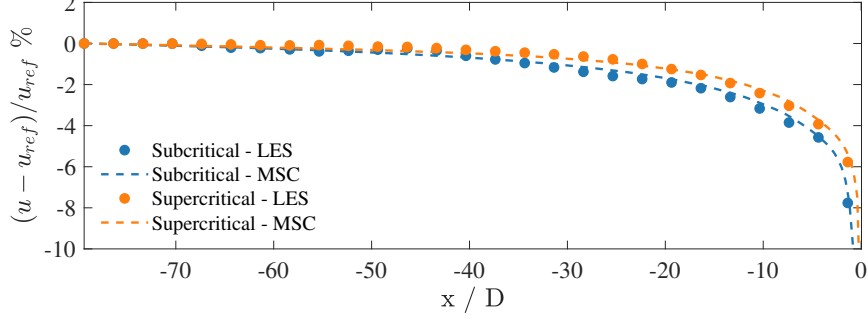

**Figure 13.** Comparison between row-averaged velocity deficit upwind of the wind farm, obtained from LES and using the MSC model, for subcritical and supercritical conditions.





Focusing on the region directly upstream the wind farm, Figure 13 displays the row-averaged percentage velocity reduction for the MSC model and LES results only, from both subcritical and supercritical conditions, highlighting the ability of the proposed model to capture the different amount of blockage produced in each case.

## 4.2 Turbine Data Validation

Regarding turbine quantities, Figure 14 shows comparisons of row-averaged thrust and power distributions between LES
and the models. First of all, looking at LES data, it can be noticed that a significant difference exists between subcritical and supercritical conditions. In the first case, pressure disturbances produced by the resonant lee waves induce velocity variations at a comparable scale to the wind farm streamwise length $L_x$. This results in a fluctuation of thrust and power that is characterized by a wavelength roughly equal to $L_x$, not observed in the supercritical regime. Interestingly, Figure 14 highlights how the wake model alone consistently overestimates both thrust and power. Moreover, results are negligibly affected when local induction
effects are added. At the same time, the wake model is indeed able to capture the global trend in power and thrust distributions, and it produces quite accurate results if diagrams are normalized by power and thrust experienced at the first row (not shown here). This highlights how absolute values are affected by a global bias that arises from failing to capture the effective velocity at the first row, i.e. the effect of large-scale gravity wave blockage. This concept of "an overprediction bias that pervades the entire wind farm", previously hypothesized by Bleeg et al. (2018), it is further corroborated by our results.

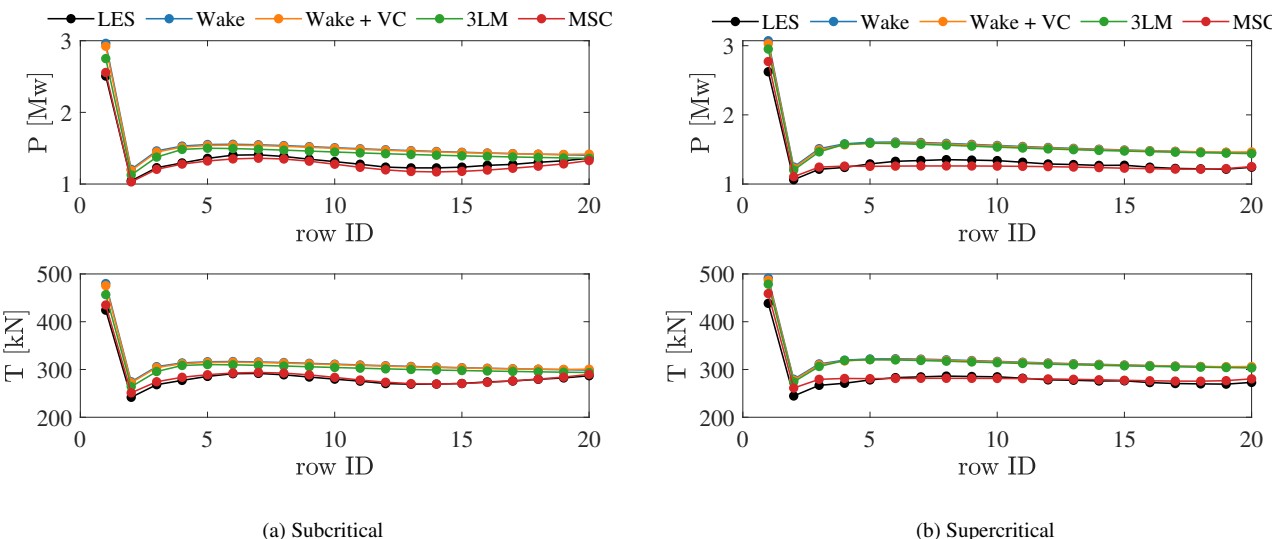

(a) Subcritical               (b) Supercritical

**Figure 14.** Comparison of row-averaged thrust and power distributions for wind farm (a) subcritical and (b) supercritical cases. Blue: wake model, orange: wake and local induction model, green: original 3LM, red: MSC model.

The original 3LM formulation of Allaerts and Meyers (2019) was the very first attempt to solve such structural deficiency of wake models. The approach does capture a certain amount of blockage, as the free stream velocity used by the wake model is uniformly reduced according to the perturbation velocity obtained from the 3LM at the upstream sampling point. We highlight





that, in this approach, the wake model is still run using a horizontally-uniform inflow velocity, thus it is not able to capture any trend inside the wind farm. As a result, thrust and power distributions are simply shifted to lower values as compared to the wakes-only approach with or without local induction. Moreover, the amount of blockage that the original 3LM is able to capture is very sensitive to the sampling location, which is clearly too far from the first row for the supercritical case. Finally, looking at results from the MSC model, it is evident how it can accurately capture both large-scale blockage effects as well as wind farm/gravity wave interactions, showing very good agreement to the LES results. A very interesting aspect to note, which is captured by the MSC model, is the combined effect of the unfavorable pressure gradient upstream the wind farm, and the favorable gradient inside. The magnitude of these effects, as well as the prevalence of one over the other, depend on many parameters such as the inversion height, strength and lapse rate. In our simulations, the first wind farm row is characterized by a lower power generation in subcritical than supercritical conditions, as gravity wave-induced blockage is higher. On the other hand, subcritical conditions lead to stronger favorable pressure gradients than supercritical inside the farm, resulting in a higher power production of waked turbines. As a result, the wind farm produces similar power for the two cases analyzed here, as shown in Table 7. This emphasizes that in addition to modeling the blockage effects, properly capturing the large-scale interactions between wind farms and the atmosphere is essential to accurately estimate wind farm power. In fact, depending on the specific stratification conditions, the combination of these effects may hamper or enhance power production as compared to fully neutral conditions.

| Total Power [MW] | LES | MSC | 3LM | Wake + VC | Wake |
|---|---|---|---|---|---|
| Subcritical | 135.0 | 131.3 | 147.6 | 153.2 | 154.0 |
| Supercritical | 133.5 | 131.4 | 156.7 | 158.3 | 159.2 |

**Table 7.** Total wind farm power obtained from LES simulations and as predicted by reduced order models.

In Figure 15 we show both the model error with respect to LES (both cases are plotted together) for each individual turbine, as well as the model error on global wind farm power for each case. Looking at the first metric (Figure 15a), it can be noticed how the wake model with and without local induction effects, as well as the original 3LM, consistently overestimate wind farm power, both at the first row (isolated points at the top-right of the diagram) and inside the wind farm, with the 3LM best performing among the three. Moreover, all these models are not able to capture the effects of wind farm gravity wave interaction on the waked turbines, resulting in a circular cloud of points shifted upwards with respect to the zero-error line. Conversely, the MSC model not only substantially improves predictions at the first row, but also captures power and thrust oscillations inside the wind farm. As a consequence, points are more distributed along the zero-error line, proving that the proposed framework can model the actual physical processes produced by thermal stratification. Regarding the second metric, i.e. global wind farm power prediction, the MSC model is substantially more accurate than all other models (2.7% and 1.6% error for the subcritical and supercritical case, respectively), additionally underestimating slightly wind farm power (a desirable effect for industrial applications). Conversely, using any of the other models always overestimates wind farm power by at least 10%, with the





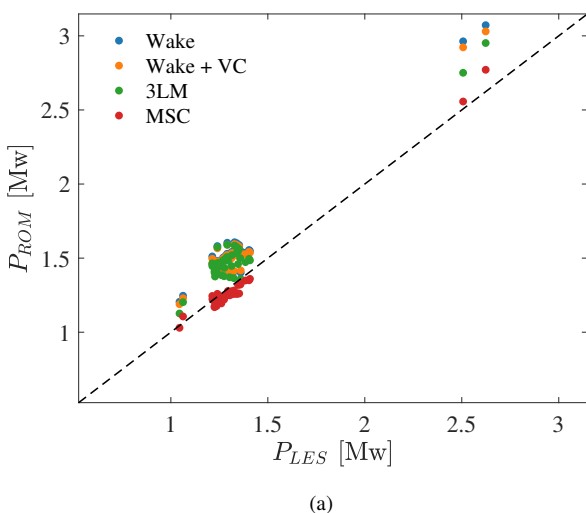
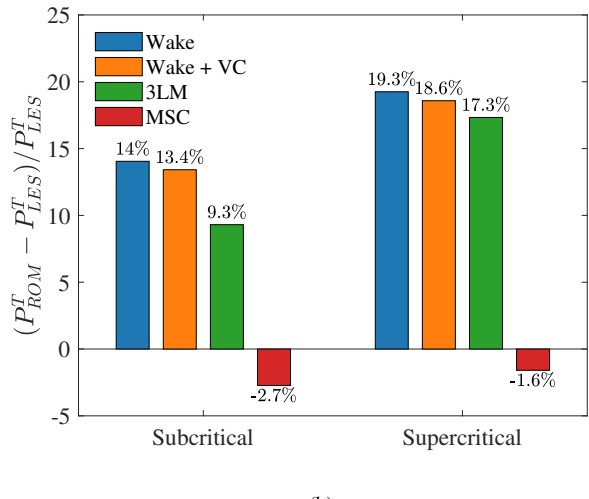

(a)             (b)

**Figure 15.** (a) Error on power for each turbine considered individually. Data from both subcritical and supercritical cases are shown together. (b) Global wind farm power error of each model in percentage of LES. Blue: wake model, orange: wake and local induction model, green: original 3LM, red: MSC model.

wakes-only approach being the least accurate (14% and 19% error for the subcritical and supercritical cases, respectively). Figure 15b also confirms that including local induction effects improves wake-model predictions only marginally.

# 5 Conclusions

The purpose of the present study was to introduce a fast, reduced-order model capable of capturing the interaction between
wind farms and the thermally stratified atmosphere, along with local induction and wake effects. We proposed the multi-scale coupled (MSC) model, which combines a heterogeneous background velocity field, computed numerically using a meso-scale model, with analytical results from wake and local induction models.

To validate the model, comparisons were made against finite wind farm large-eddy simulations (LES) characterized by similar velocity and shear stress profiles but different potential temperature structures. Due to the high amount of computational
resources required to conduct the above LES studies, model validation has only been performed against two atmospheric states. Specifically, while a neutral stratification within the ABL and a lapse rate of 1 K/km have been prescribed in both cases, the inversion strength has been varied to produce a subcritical regime, where interfacial waves can propagate upstream, and a supercritical regime, where upstream propagation is only performed by internal waves. The background atmospheric state for both the original three-layer and MSC models have been computed from LES results, while turbine thrust and power curves
have been evaluated by running several non-turbulent isolated turbine simulations. Furthermore, the turbulence intensity model used in the wake model was adjusted to match the turbulence intensity observed in LES.





The LES results revealed substantial differences in the underlying physics between the two regimes, despite observing a similar total wind farm power. In the subcritical case, characterized by a higher blockage and lower first-row power, gravity waves induced a stronger favorable pressure gradient within the wind farm compared to the supercritical case. Consequently,

the waked turbines in the subcritical regime experienced a faster wake recovery. These opposing effects led to comparable overall wind farm power in both cases.

Regarding the MSC model, we found that the 3LM step strongly over-predicts the depth-averaged velocity deficit in the cluster wake. We believe that this is due to a poor modeling of the shear stress terms within the 3LM, suggesting that further research is required in this direction. Nevertheless, this issue has negligible impact on the background velocity field resulting

from the reconstruction step. In fact, only the large-scale perturbation pressure, accurately predicted by the 3LM, is used in the 3LMR step to force the three-layer equations. In addition, comparisons of wind turbine thrust, power, and wind velocities demonstrated the MSC model's ability to accurately capture the intricate interaction between wind farms and the atmosphere. In contrast, the wake-only approach, with or without local induction effects, failed to differentiate between the two regimes and produced results solely dependent on the background velocity profile, disregarding global blockage effects. This introduced

a positive wind speed bias that propagated downstream, resulting in an overestimation of power throughout the wind farm. Although the original 3LM formulation partially addressed this issue by correcting the free stream velocity used in the wake model, it failed to capture the beneficial pressure gradient effect within the wind farm due to the uniform and non-local velocity correction. Nevertheless, the model still overestimated power within the wind farm, particularly in supercritical conditions where the wind correction became minimal due to the large upstream sampling distance of 10 rotor diameters, leading to an

underestimation of blockage effects.

Overall, if compared against LES, the MSC model underestimates wind farm power by $\approx 2\%$, while all other approaches overestimate power from $\approx 10\%$ to $\approx 19\%$, with the original 3LM and the wake only approach without local induction being the most and least accurate respectively.

We believe that the coupling framework introduced in the MSC model could present a significant contribution towards

advancing our understanding of the complex physical phenomena arising in large-scale wind farms. Additionally, it may help refine the power estimation methodologies for existing and future wind farm clusters, enhancing the accuracy and reliability of power predictions.



## Appendix A: Near Wake Correction for Gaussian Wake Model

As pointed out in Section 2.1.2, the Bastankhah and Porté-Agel (2014) Gaussian wake model is limited in predicting wake deficit in the near wake. In particular, Equation (9) becomes singular when $C_T > 8\sigma^2/D^2$. Hence, based on the value of $C_T$, the singularity region may extend more or less behind the rotor, or be non existent at all. Nevertheless, as values of $C_T$ in below-rated turbine operation conditions range from 0.7 to 0.9, a singularity region is to be expected in many cases. Moreover, since wake width $\sigma$ also depends on $C_T$ through Equation (10) in a non-linear way, the specific distance at which Equation (9) turns non-singular cannot be evaluated unless a numerical method is used and, additionally, $C(x)$ would be zero at such location, which is of course non physical.

To overcome this set of issues, we propose a near wake correction for the Gaussian model, by smoothly transitioning from the super-Gaussian wake model of Shapiro et al. (2019b) to the Bastankhah and Porté-Agel (2014) profile in the far wake through a damping function. The resulting model is fully equivalent to the Gaussian model at downstream distances greater than 4 diameters, while it approaches a top-hat distribution close to the rotor. In addition to the latter property of the super-Gaussian wake model, the resulting deficit function possesses the desirable feature that the maximum deficit is not located directly past the rotor disk, but somewhere in the near wake (Blondel and Cathelain, 2020).

According to Shapiro et al. (2019a), the functions $f(r,\sigma)$ and $C(x)$ in the super-Gaussian wake model are given by

$$f(r,\sigma) = \exp\left(-\frac{r^n}{2\sigma^2}D^{2-n}\right) \tag{A1}$$

$$C(x) = 2^{2/n-1} - \sqrt{2^{4/n-2} - \frac{nC_T}{16\Gamma(2/n)(\sigma/D)^{4/n}}} \tag{A2}$$

where $\Gamma$ is the gamma-function. The coefficient $n$, which is a function of the downstream distance from the rotor, determines the deficit shape. When $n = 2$, the model is fully equivalent to the Gaussian wake model, while $C(x)$ tends to a top-hat distribution for higher values of $n$. To enhance the performance of the model in the near wake, we evaluate $n$ as follows

$$n(x) = w_n(x)n_n(x) + w_f(x)n_f(x) \tag{A3}$$

$$n_n(x) = 2\exp(-0.68x/D) + 2 \tag{A4}$$

$$n_f(x) = 2 \tag{A5}$$

$$w_n(x) = \frac{1}{2}\left[1 - \tanh\left(7(x - x_c)/\delta\right)\right] \tag{A6}$$

$$w_f(x) = \frac{1}{2}\left[1 + \tanh\left(7(x - x_c)/\delta\right)\right] \tag{A7}$$

where $n_n$ and $w_n$ are the exponent and weight associated to the near wake, while $n_f$ and $w_f$ pertain to the far wake. The parameters $x_c$ and $\delta$ are set to $2D$ and $4D$ respectively. Figure A1 shows the evolution of $n_n$, $n_f$ and $n$, as well as the weights associated with the near and far wake regions. As can be noticed, $n$ has already fell down to 2 at 4 downstream diameters, while it increases approaching the rotor so that wake deficit tends to a top-hat distribution. On the other hand, the damping functions associated with the weights cross at $2D$, and are characterized by a merging region of $4D$, which is controlled by parameters $x_c$ and $\delta$.





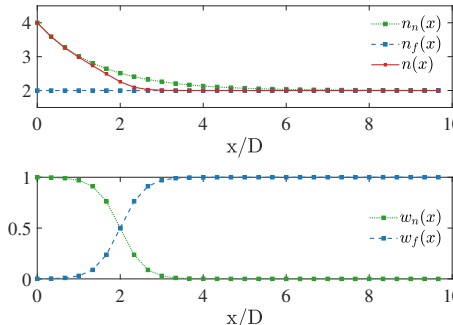

**Figure A1.** (top) streamwise evolution of overall, near and far wake components $n$, $n_n$ and $n_f$ respectively; (bottom) near and far wake weighting function evolution.

Figure A2 compares velocity evolution along the wake centerline, for the super-Gaussian and Gaussian models, the latter with and without the proposed near wake correction. On the left (Figure A2a) the whole velocity field is shown for a free stream wind speed $U_\infty$ of 8 m/s and a $C_T$ coefficient of 0.9 while, on the right (Figure A2b) we show velocity evolution at the wake centerline for three different values of $C_T$, at the same wind speed. As can be noticed, the Gaussian model is singular below roughly $1.5D$, for $C_T = 0.9$, whereas a null wind speed is reached slightly below $1D$ with $C_T = 0.6$. For the super-Gaussian model we used the coefficients proposed by Blondel and Cathelain (2020). These lead to a stronger deficit in the far wake region, and a minimum wind speed located around $2.5D$ in all cases.

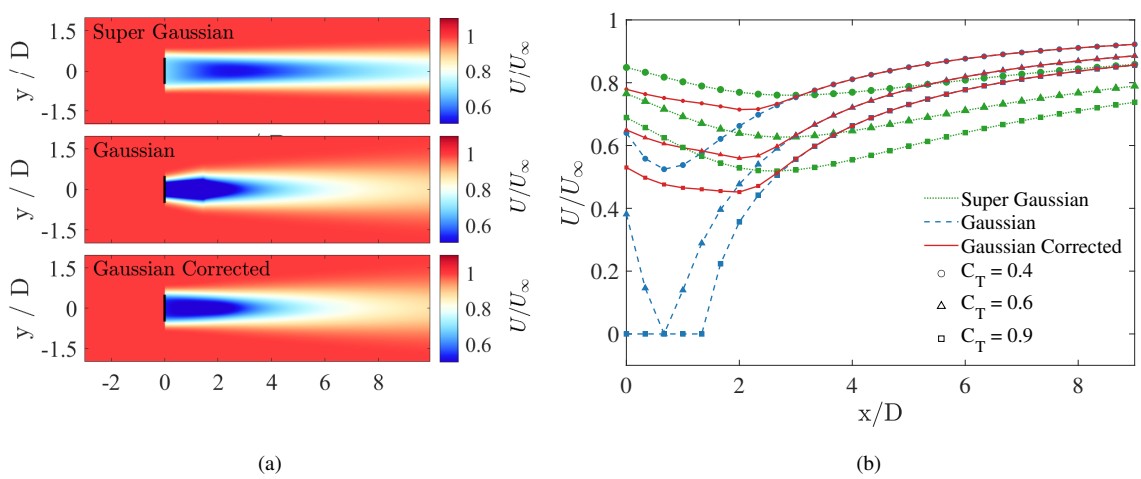

**Figure A2.** (a) flow field at the hub-height predicted by the super-Gaussian, Gaussian and Gaussian with near wake correction, for a wind speed of 8 m/s and a thrust coefficient of 0.9; (b) streamwise evolution of the wind velocity at the wake centerline for the three models, for three different values of the thrust coefficient. The singularity region of the Gaussian wake model is evident from the diagrams.



The Gaussian model with near wake correction predicts a slightly lower velocity in the near wake than the super-Gaussian model, while the maximum deficit is located around $2D$. After $4D$, the predicted velocity exactly overlaps data obtained with the original Gaussian wake model, highlighting how the two produce equivalent results when turbine spacing is greater than such distance.

## Appendix B: Turbine Data for the NREL 5MW

To model wind turbines in our LES simulations, we use an actuator disk model that takes into account the detailed rotor information instead of just using prescribed turbine $C_T$ from curves, that are used instead to evaluate thrust and power in reduced order models. This introduces an extra source of uncertainty in comparing low and high fidelity data, as a power or thrust differences could be both due to structural model deficiencies or simply to the fact that different turbine $C_T$ or $C_P$ are used for a given wind speed. Moreover, for the NREL 5MW turbine, a reliable $C_T$ vs wind speed curve is not available in the literature. For instance, the thrust curve provided in Jonkman et al. (2009) also contains rotor inertial forces, resulting in a $C_T$ vs wind speed curve that is biased towards higher values.

These considerations motivated us to evaluate $C_T$ and $C_P$ curves for the NREL 5MW turbine using LES, thus removing any source of bias when comparing reduced-order model results against high-fidelity data from LES. Specifically, we run 16 isolated turbine simulations, spanning wind speed magnitudes from 5 to 20 m/s, using the same turbine model adopted for the wind farm simulations. The inflow profile is characterized by a uniform velocity, while the outlet boundary condition is set to zero gradient on all fields. Top and bottom, as well as spanwise boundaries are set to periodic. The domain extends 10D from the rotor center in all directions and has a uniform spacing of 10 m. The simulations have been advanced in time for 600 s, and data have been averaged for the last 300 s. The Gaussian projection width has been set to 18.75 m.

Figure B1 shows turbine quantities regulated by the control system, i.e. rotor omega, blade pitch and generator torque, alongside the resulting $C_T$ and $C_P$ curves.

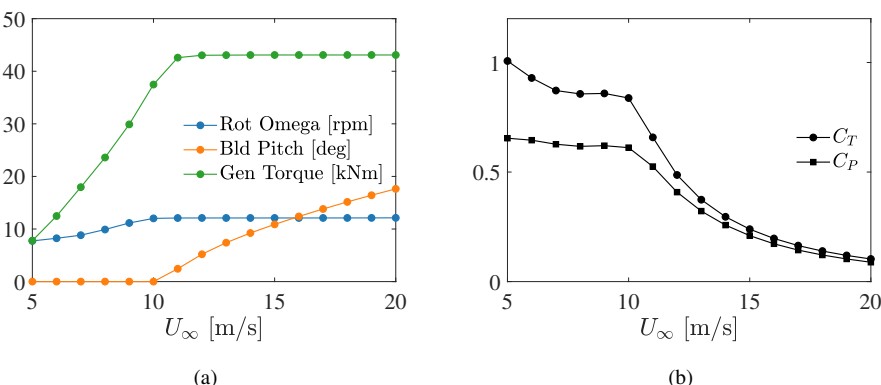

**Figure B1.** (a) rotor omega, blade pitch and generator torque as a function of wind speed, resulting from the control system action; (b) thrust and power coefficient curves as a function of wind velocity.





Data from Figure B1b have been used to update $C_T$ and $C_P$ values at each MSC model coupling iteration, as well as in all other reduced model runs reported in the present paper.

### Appendix C: 3LM/3LMR Solution

In order to solve Equations (1), (2) and (5), we first transform them in Fourier space. This leads to a $7 \times 7$ linear system for each $k$ and $l$ wavenumber couple, with $k$ and $l$ the wavenumbers associated with the $x$ and $y$ directions, respectively. After some manipulation, displacements $\eta_1$ and $\eta_2$ can be eliminated from the equations, and pressure can be directly expressed as a function of the perturbation velocity components in the wind farm and upper layers, reducing the number of equations to 5 for each wavenumber couple. To speed up the numerical solution, linear systems associated to each wavenumber couple are

combined to form a unique system, characterized by a block-diagonal solution matrix $A^{3LM}$ of size $5N_kN_l \times 5N_kN_l$, where $N_k$ and $N_l$ are the wavenumber sizes in the streamwise and spanwise directions respectively. We use a sparse matrix format to express $A^{3LM}$, while the complex linear system is solved using the generalized minimum residual (GMRES) (Saad and Schultz, 1986) iterative method. The full linear system, corresponding to the 3LM solution, is given by

$$
\begin{bmatrix}
A(k_1,l_1) & & & \\
& A(k_1,l_2) & & \\
& & \ddots & \\
& & & A(k_{Nk},l_{Nl})
\end{bmatrix}
\begin{bmatrix}
s(k_1,l_1) \\
s(k_1,l_2) \\
\vdots \\
s(k_{Nk},l_{Nl})
\end{bmatrix}
=
\begin{bmatrix}
b(k_1,l_1) \\
b(k_1,l_2) \\
\vdots \\
b(k_{Nk},l_{Nl})
\end{bmatrix},
\tag{C1}
$$

where

$$
A(k_i,l_j) =
\begin{bmatrix}
i\sigma_1 + \nu_{t,1}|\kappa|^2 + (D_{11}+C_{11})/H_1 & -D_{11}/H_1 \\
f_c + (D_{21}+C_{21})/H_1 & -D_{21}/H_1 \\
-D_{11}/H_2 & i\sigma_2 + \nu_{t,2}|\kappa|^2 + D_{11}/H_2 \\
-D_{21}/H_2 & f_c + D_{21}/H_2 \\
k_i H_1 \Phi/\sigma_1 & k_i H_2 \Phi/\sigma_2
\end{bmatrix}
$$

$$
\begin{matrix}
-f_c + (D_{12}+C_{12})/H_1 & -D_{12}/H_1 & ik_i/\rho \\
i\sigma_1 + \nu_{t_1}|\kappa|^2 + (D_{22}+C_{22})/H_1 & -D_{22}/H_1 & ik_i/\rho \\
-D_{12}/H_2 & -f_c + D_{12}/H_2 & ik_i/\rho \\
-D_{22}/H_2 & i\sigma_2 + \nu_{t,2}|\kappa|^2 + D_{22}/H_2 & ik_i/\rho \\
l_j H_1 \Phi/\sigma_1 & l_j H_2 \Phi/\sigma_2 & 1/\rho
\end{matrix}
\tag{C2}
$$





and

$$
\quad s(k_i, l_j) = \begin{bmatrix} \hat{u}_1(k_i, l_j) \\ \hat{u}_2(k_i, l_j) \\ \hat{v}_1(k_i, l_j) \\ \hat{v}_2(k_i, l_j) \\ \hat{p}(k_i, l_j) \end{bmatrix}, \quad b(k_i, l_j) = \begin{bmatrix} \hat{f}_x(k_i, l_j) \\ \hat{f}_y(k_i, l_j) \\ 0 \\ 0 \\ 0 \end{bmatrix}. \tag{C3}
$$

In the above equations, $\hat{f}_x$ and $\hat{f}_y$ are the Fourier transforms of the Gaussian-filtered wind farm force defined in Equation (18), $\Phi$ is the complex stratification coefficient, defined as the complex quantity between square brackets in Equation (5), $|\kappa|^2$ is equal to $(k_i^2 + l_j^2)$, while $\sigma_1$ and $\sigma_2$ can be evaluated as $(k_i U_1 + l_j V_1)$ and $(k_i U_2 + l_j V_2)$ respectively, and are referred to as the intrinsic frequencies in the wind farm and upper layers (Smith, 2010).

For the 3LMR step, i.e. Equations (21) and (22), the procedure is identical upon elimination of the last row and column from $A(k_i, l_j)$. By doing so, the size of $A^{3LMR}$ becomes $4N_k N_l \times 4N_k N_l$. The unknown and right hand side of the linear system associated with the 3LMR step are defined as

$$
s(k_i, l_j) = \begin{bmatrix} \hat{u}_1^{\mathrm{bk}}(k_i, l_j) \\ \hat{u}_2^{\mathrm{bk}}(k_i, l_j) \\ \hat{v}_1^{\mathrm{bk}}(k_i, l_j) \\ \hat{v}_2^{\mathrm{bk}}(k_i, l_j) \end{bmatrix}, \quad b(k_i, l_j) = \begin{bmatrix} -ik_i \hat{p}(k_i, l_j)/\rho \\ -il_j \hat{p}(k_i, l_j)/\rho \\ -ik_i \hat{p}(k_i, l_j)/\rho \\ -il_j \hat{p}(k_i, l_j)/\rho \end{bmatrix}. \tag{C4}
$$

After solving the two linear system associated with matrices $A^{3LM}$ and $A^{3LMR}$ in cascade, $\hat{\mathbf{u}}_1^{\mathrm{bk}}$ components are transformed

back into physical space, and Equations (25) to (27) are used to reconstruct the background velocity field $\mathbf{U}_b(\mathbf{x})$. The latter is then interpolated at turbine quadrature points from the 3LM grid by means of bi-linear interpolation formulas.



*Code availability.* TOSCA is available at https://github.com/sebastipa/TOSCA

*Author contributions.* Conceptualization, S.S, D.A., J.B.; methodology, S.S.; software, S.S, A.A.; validation, S.S.; formal analysis, S.S.; investigation, S.S.; computational resources, J.B.; data curation, S.S.; writing–original draft preparation, S.S.; writing–review and editing, J.B., D.A.; visualization, S.S.; supervision, J.B., D.A.; project administration, J.B.; funding acquisition, J.B.. All authors have read and agreed to the published version of the manuscript.

*Competing interests.* No competing interests are present.

*Acknowledgements.* The present study is supported by UL Renewables and the Natural Science and Engineering Research Council of Canada (NSERC) through Alliance grant no. 556326. Computational resources provided by the Digital Research Alliance of Canada (www.alliancecan.ca) and Advanced Research Computing at the University of British Columbia (www.arc.ubc.ca) are gratefully acknowledged.



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
