# Peer review of "The Multi-Scale Coupled Model: a New Framework Capturing Wind Farm-Atmosphere Interaction and Global Blockage Effects"

_Wind Energy Science, 2023_

## Author Comment (AC1)

**University of British Columbia**

**UBCO-UL NSERC Alliance Grant "Reduced-Order Models of Wind Farm Induction and Far-Field Wake Recovery"**

**Response to Reviewer 2**

Exec. S. Stipa - November 14, 2023

We would like to thank the reviewer for the time dedicated to revising the paper. We proceed with answering and clarifying, where possible, the proposed comments.

Our response, denoted in black, is shown below. Modified text of the paper is shown between quotes in *italic*, while the reviewers' comments are denoted in blue. Please refer to the track changes section at the end of this document for a detailed overview of the changes made to the manuscript.

This is an excellent, highly relevant contribution to the field of wind farm atmosphere interaction, presenting a multiscale coupled model framework, coupling models that resolve effects from turbine scale to meso-scale. The paper is well written, and the authors provide a very clear description of the models and how they are coupled. The novelty of the contribution is the three-layer model reconstruction (3LMR), the background velocity reconstruction and the new wake superposition.

The MSC model is verified against results from LES simulations for two cases with neutral conditions in the boundary layer and stable conditions above, one supercritical and one subcritical.

The ability of the MSC to better capture the LES results than the earlier 3 layer model is evident. The MSC (and the LES it is verified against) clearly demonstrate the significance of the stability conditions above the boundary layer for the magnitude of the wind farm blockage.

While the verification against a LES simulation is a very valuable exercise, I would recommend as further work (not for the current paper, but as a follow up project to the current submission) some investigations comparing the model against field data (SCADA data as well as dual scanning lidar data measuring blockage upstream of wind farms). Once validating against field data, binning measurements for finite sectors, and averaging the model results over a range of directions, it is possible that some of the conclusions might change slightly, since derived from simulations for a single direction aligned with the wind farm layout.

We sincerely appreciate the reviewer's comments on the paper. We fully agree with the reviewer's recommendation to compare the MSC model against field measurements. Currently, we implemented the MSC model in OpenWind® (this research is partially enabled by UL Renewables) and an effort is underway to validate the model against observations gathered in the GloBE project (Adams et al., 2023). Moreover, we changed the word validation to verification throughout the paper, as we are dealing with comparison against LES rather than observations.

For the current paper, I would also recommend to clarify that the magnitude of the effects reported, which imply large reduction in wind farm output, are only applicable for the conditions simulated, rather than over the typical overall conditions that will be experienced by a wind farm.

The reviewer suggestion has been implemented in the paper.

The MSC model can potentially deliver increased fidelity compared to standard engineering models for wakes and blockage, at the fraction of the cost of more expensive high-fidelity simulations such as RANS CFD or LES. How useful the model will end up being for the wind industry will depend on the ability for the users to source the many input parameters that the model requires. Inputs such as the potential temperature profile, the eddy viscosity profiles, the background wind angle profile, etc.. are typically not measured up to the heights of interest (i.e. through the boundary layer height). Have the authors already considered if the outputs from meso-scale models or re-analysis data sets could provide the required information?

Yes, both our research version of the model and the OpenWind implementation can now use reanalysis data to set the model inputs for the potential temperature profile. Using ERA5 renalysis data has already

been done by Allaerts et al. (2018) to assess blockage effects on annual energy production for the 3LM. Mesoscale models are another viable option. Text explaining this has been added in the paper.

Abstract: it would be a good idea to clarify that the 'overestimation of the wind farm power by 13% to 20% is seen for the conditions modelled ( 9 m/s at hub height, therefore high Ct, and also for very specific stability conditions), rather than over the whole range of site conditions.

The abstract has been rephrased according to the reviewer's suggestions.

28: 'cannot be described by a simple combination of individual wake deficits' . Should this be softened a bit? Empirical wake models are doing reasonably ok on the wind farm sizes onto which they have been calibrated. May be say that models combining individual wake deficits have their limitations, and explain what these are (e.g. need for a wake superposition model, response to local changes in wind speed/turbulence intensity/shear/veer when these fall outside of the range within which the models have been typically validated).

The literature review section mentioned by the reviewer has been rephrased.

33: when you write 'Regarding blockage, or induction (Bleeg et al., 2018)', this kind of suggests that Bleeg et al imply that blockage is synonym with induction. I don't think this is the case. Might be better to add the reference to Bleeg et al at the end of the sentence. Because in the Bleeg et al results, the interaction between the wind farm and the thermally stratified atmosphere is very much playing a role in the magnitude of the wind farm blockage.

The reviewer is right. The meaning of induction is more related to momentum theory rather than thermal stratification, hence this word has been removed from the sentence.

52: 'reducing the free stream velocity'. Always reducing? I'd expect mesoscale effects for offshore wind farms operating downstream of a coastal transition to see acceleration of the freestream flow towards the back of the wind farm. Or possibly expect lateral gradients in the background flow if the fetch to the coast varies for different parts of the wind farm. Wouldn't you?

May be this is just because the 'meso-scale effects' accounted for by the 3LM are only those representing the feedback from the wind farm onto the background flow. If so , might it be worth clarifying?

The reviewer is right in general, but in this specific case we are referring to the correction provided by the 3LM to the freestream velocity used by the wake model. In the original coupling between the 3LM and the wake model, the 3LM solution is sampled upstream of the wind farm cluster, for this reason the wind farm power predicted by the model is always reduced w.r.t to the wakes-only formulation. This is exactly the reason why such coupling is limited, i.e. [it] "fails to adequately transfer the influence of the large-scale physical processes to the flow at the turbine scale". This includes background flow acceleration towards the end of the wind farm or – not treated in our paper – wind speed up due to coastal gradients. Although we rephrased the section mentioned by the reviewer to highlight the first aspect, we feel that the discussion about coast proximity falls outside of this specific discussion about the limitation of the original 3LM coupling.

54: 'the coupling between the turbine-scale wake effects and the meso-scale global effects is weak...': is this documented somewhere? If so, please provide a reference.

This is explained more in detail when we go over the original coupling technique in Sec. 2.1.5, but no specific reference addressed this aspect. Nevertheless, this can be easily seen based on how the coupling

is defined in Allaerts and Meyers (2019). Specifically, after the 2D spatial solution is obtained at the mesoscale, only one point of this solution (10 diameters upstream the wind farm) is used to update the freestream velocity for the wake model. This is clearly limited as the new velocity is just shifted uniformly. Conversely, in our approach, a different freestream velocity is chosen at each turbine location, interpolating the mesoscale field. This allows to transfer large scale velocity gradient at the microscale, capturing the flow deceleration upstream, but also the beneficial/detrimental effects inside the wind farm.

74-77: 'In the subcritical regime ... due to internal waves'. Does this belong to the discussion/conclusion section?

We rephrased the sentence without drawing any conclusions, but just explaining the differences between the two regimes from a physical standpoint.

118: Questions about the three layers structure: Wind turbines are getting taller... assume a 15MW Vestas turbine, rotor diameter of 240m with a hub height of 2/3 the RD, has a tip height of 280m. Offshore boundary layer heights can be quite low. How high do you assume the geostrophic height to be above the turbine layer in the model? Could you please comment if this is compatible with realistic offshore conditions?

The turbine top tip piercing the inversion layer is a situation in contrast with the perturbation analysis used to develop the mesoscale model. This is an extreme case that has been studied through LES in Lanzilao and Meyers (2023), and would probably require ad-hoc corrections to the mesoscale model (we did not investigate such condition). However, we analyzed the statistics of the inversion height e.g. in the North See using ERA5 reanalysis data. For instance, Lanzilao and Meyers, 2023 did the same and found a value of around 200 m, but they fitted ERA5 data with a temperature model that could only provide neutral conditions below the ABL height, which in many cases provides fits that are very far from the actual profile. In our fit, we used a temperature model that could distinguish between convective, neutral and stable conditions (the most frequent), and we found that on average the inversion height is located around 500 m (Fig. 1).

[Figure]

**Figure 1.** Statistics of parameters defining the potential temperature profile at the location of the N4 cluster in the North Sea, extracted from hourly-averaged ERA5 reanalysis data of years 2021-2022. Variables are as follows: $\gamma$ [K/km] is the lapse rate, $\gamma_{ABL}$ [K/km] is the lapse rate below the inversion layer, $\Delta T$ [K] is the potential temperature jump across the inversion layer, $\Delta h$ [km] is the inversion width, $H$ [km] is the inversion height. The profile used to fit the data is continuous, piecewise linear with three sections, i.e. the region below the inversion layer (unstable, neutral or stable), the inversion layer and the free atmosphere (always stable, at most neutral)

.

In these conditions, as demonstrated in our paper, results are pretty accurate. When the ABL height

decreases, the main problem is that the second layer starts to disappear, as the height of the first layer is fixed to twice the hub height. In this case – not included in the present study – we analyzed a modification to the model where we relax the definition of $H_1$ below a certain value of $H$ so that it is defined as half of the ABL height (this leads to $H_2 = H_1$), and we distribute the thrust force also to the second layer based on the rotor disk area fraction that is above $H_1$). Although still under investigation, this does not seem to show any numerical discontinuities when transitioning to the layer height definitions presented in the paper. In this situation, the ABL is shallow enough to remove the need of a distinction between the upper layer and the wind farm layer.

How are H1 and H2 set/defined?

In the paper, this quantities are defined in Sec. 2.3 "Background State".

141: where is the lapse rate defined? Lapse rate above the geostrophic level?

Yes, this is the free atmosphere lapse rate, it has now been specified. The definition of all input parameters is explained in 2.3 and their numeric values are summarized in Sec. 4 (Table 6).

142: 'dtetha is the potential temperature jump across the inversion layer': how thick is the inversion layer? Is the thickness affecting the model results?

The inversion layer is modeled as a discontinuity in potential temperature, as we use shallow water wave theory to model interface waves. This approximation is consistent with previous studies such as Allaerts and Meyers (2019); Smith (2010). In reality and in our LES simulations, interface waves are in reality trapped waves within the inversion layer. This approximation is sufficiently accurate for reasonably thin inversion layer, such that it does not become part of the ABL. As mentioned, in the LES simulations we used an inversion thickness of 100 m and the vertical profile of potential temperature given by Rampanelli and Zardi (2004), and the gravity waves patterns agree well.

151: 'zero vertical pressure gradient inside the ABL': is this limiting the applicability of the model to neutral conditions in the boundary layer?
In principle, yes. However, we are planning to extend the 3LM to formally account for internal ABL stability other than neutral by modifying the parametrization of the tensors $C_{ij}$ and $D_{ij}$. Moreover, the mesoscale model can already partially account for some effects produced by internal ABL stability, as they would change the background input. In particular, the layer-averaged velocity would change ($\mathbf{U}_1$, $\mathbf{U}_2$ and $\mathbf{U}_g$), together with the shear stress at $H_1$, which would be reduced, in turn changing $C_{ij}$ and $D_{ij}$. For the coupling strategy, stability should be accounted for in the velocity reconstruction as mentioned in the dedicated section. However, the capability that the MSC model already possesses to capture stability effects has not been yet assessed.

152: 'p can only change in response to a vertical displacement of the ABL' : Shouldn't p also respond to the presence of the wind farm, in that the horizontal p gradient is usually linked to the Coriolis term. If the wind farm removes momentum it also changes the balance between Coriolis and pressure gradient. If the pressure in the model is not responding to this change, does it mean that the model might produce some flow acceleration/deceleration which are not correct?

When deriving the 3LM perturbation equations (Allaerts and Meyers, 2019), the fact that geostrophic wind (i.e. the mean pressure gradient), vertical momentum transport and Coriolis force in the unperturbed state are in balance is exploited to eliminate these terms from the momentum equations, leaving only their perturbations. However, the mesoscale driving pressure gradient should not feel the presence of the wind farm, as it is on a much larger scale, hence it is not perturbed (this is equivalent to saying that the wind farm

does not perturb the geostrophic wind). This is consistent with the LES simulations, where the background pressure gradient that is applied in the successor simulation is the same as the precursor. Instead, velocity variations produce an imbalance in the geostrophic equilibrium by varying Coriolis force and vertical momentum transport, and this is modeled in the mesoscale model through the Coriolis and $C_{ij}/D_{ij}$ tensors.

Or does this statement only apply to the pressure boundary condition that is the third layer?

The pressure is the same throughout the ABL, while the third layer is in reality just a boundary condition that links the inversion displacement to the pressure perturbations sent back into the ABL from the free atmosphere response.

175: 'at wind turbine locations'. Should this be 'at downstream wind turbine locations'?

No, this is to distinguish between using the wake model to extract power/thrust (i.e. computing velocity only at turbine locations) or to evaluate velocity on a uniform grid, i.e. at every location on a plane. Here, some query points are located in the undetermined region as shown in Fig. 12 and 13, where we look at the entire velocity field. Conversely, wake models are usually employed to extract power/thrust (e.g. Fig. 15), which is faster as they only require evaluation at the turbine location or on a few disk locations if using quadrature points, and it does not require their evaluation in the near wake.

312: 'they do not account for turbine-ABL interaction'. I don't know that this statement is completely true... a lot of wake models, initially tuned on capturing single wakes, were re-tuned based on e.g. capturing PoP at wind farms such as Horns Rev/Nysted/Rodsand... rather than tuned against LES results... so, while they don't explicitly account for the turbine = ABL interaction, they implicitly account for some of it through their calibration.

The reviewer correctly points out that engineering model tuning allows to indirectly include physical processes and effects not directly captured in their formulation. The sentence has been rephrased to clarify this concept.

Equations 21 & 22: how is mass conservation between layer 1 and 2 satisfied in the reconstruction step?

The background velocity cannot satisfy mass conservation. In fact, this velocity is the result of the perturbation pressure field obtained when assuming the same boundary layer displacement as if the wind farm was present. The mass conservation equations contain an imbalance due to the velocity deficit directly produced by the removal of momentum from the wind farm. Specifically, the 3LM solution can be decomposed into background velocity + wind farm wake which, together, satisfy mass conservation across layers.

343: given that you include Coriolis in your model equations, how valid is your assumed inflow profile? (thinking about tall turbines that might reach into the Ekman layer).

The inflow profile is decomposed in magnitude and direction. For the magnitude, we use the classic similarity laws in the surface layer, while the angle is calculated from the LES precursor simulations. However, in absence of that, we propose other methods in Sec. 2.3 to compute the inflow angle and/or magnitude, such as ERA5 reanalysis data or the Nieuwstadt (1983) model, both featuring wind veer (in particular both input modes are currently implemented in both our research and OpenWind® versions of the MSC model). We found the Nieuwstadt (1983) model to be pretty accurate regarding the wind angle if compared with LES results. Conversely, velocity magnitude can differ sightly, and this may have a significant impact especially close to $F_r \approx 1$, where a small change in velocity could make the flow either subcritical or supercritical. In the context of the present study, where we validate the model against LES, we wanted to remove any potential error associated with the input background state (hence not strictly related

to the model accuracy itself). For this reason we decided to use the LES data to define the background inputs of Tab. 6.

Is the background shear stress magnitude at the wall related to the friction velocity and the density? (i.e. if you have two of these as inputs, is the third one an input parameter too?)
In the context of our paper, following Allaerts and Meyers (2019) we identify $\tau_{xz} = \overline{u'w'}$ and $\tau_{yz} = \overline{v'w'}$, while $|\tau| = \sqrt{\tau_{xz}^2 + \tau_{yz}^2}$, so this is not the shear stress that the reviewer has in mind, but rather its ratio with the constant density $\rho_0$ (this can be seen from the specified unit of measure in Tab. 1). Apart from this yes, $|\tau| = u*^2$ at the wall, but the two inputs have been left in Tab. 1 for completeness. To highlight this aspect we added a sentence that specifies the above condition.

What height is $TI_\infty$ taken (hub height?)

Yes, the hub height. We have clarified this.

Section 3.1: the description of the LES methodology, with the use of precursor, successor simulations and the role of the fringe region could be more clearly explained.
The reviewer raises a fair point, but the paper is already very lengthy. We covered these aspect with extreme detail in our previous paper Stipa et al. (2023), that is accessible for the interested reader (the subcritical case is the same described there, while the supercritical case is identical except for a different value of the inversion strength). There, TOSCA's methodology is explained (fringe region, controller, hybrid off-line/concurrent precursor and geostrophic damping), together with a step-by-step description on how the simulation has been carried out. We believe that it would be both lengthy and repetitive to include these detailed aspects in the present paper.

473: 'we use both periodic boundary conditions and a fringe region located at the domain inlet': Do you mean a fringe region at the inflow has periodic BCs, and the resulting profile is applied at an inflow BC to the simulation domain downstream of the fringe region (as opposed to there being periodic BCs at the inflow and outflow downstream of the windfarm?

Maybe a schematic showing the respective domains would help, illustrating the quantities listed in Table 2 and 3.

The sentence literally means that the successor simulation uses periodic boundary conditions at the inlet and outlet, combined with a fringe region located at the inlet. This is nothing more than a Rayleigh damping layer located at the inlet that extends for some distance into the domain, where instead of forcing the flow to a reference velocity imposed by the user (in the case of Rayleigh damping the vertical velocity component is forced to zero at the domain top, where the low is laminar) the flow is forced using a time and spatially resolved field, available at each time step, obtained from a precursor simulation that runs simultaneously with the successor, i.e. the concurrent precursor. This simulation in turn is initialized with inflow-outflow boundary conditions using a mapped inflow boundary from an off-line precursor run on a smaller domain. Once the turbulent solution has filled the concurrent precursor domain, we switch also its boundary conditions to streamwise periodic, and the simulation is now self-sustained, i.e. it does not require any external turbulence mapping. An entire section of our previous paper (Stipa et al., 2023, where also a schematic is shown) details this procedure, and explains the reason why we employ such method, namely to prescribe an unperturbed turbulent inflow to the successor (the wind farm wake re-advected into the domain by periodic boundary conditions is canceled out within the fringe region length) at the same time damping gravity waves reflections from the inlet and outlet boundaries that would otherwise pollute the LES solution. Other studies use a similar set-up (see for example Allaerts and Meyers (2017, 2018); Lanzilao and Meyers (2022, 2023)).

493: 'conducted on a domain smaller than the fringe region': how large was this domain? Large enough to avoid artificially increasing the correlation between developing turbulence structure (from what I presume are periodic BCs).

This part of the simulation (i.e. the off-line precursor run) and its results are described in Sec. 3.2. In particular, "The domain size of the two CNBL simulations is of 6 km × 3 km × 1 km in the streamwise, spanwise, and vertical directions respectively. The mesh has a horizontal resolution of 15 m, while in the vertical direction it is graded equally as the concurrent precursor and successor simulations." This domain is larger or in-line with previous literature (Calaf et al., 2010; Churchfield et al., 2012).

495: 'Inflow slices are then periodized along the spanwise direction and mapped at the concurrent precursor inlet, as it uses inflow-outflow boundary conditions in this initial phase'. Not sure what this means.
The concurrent precursor domain is larger than the off-line precursor domain. In particular, it is equivalent to the successor in the spanwise and vertical directions, which are mainly dictated by the ability to capture gravity waves. Hence, as the concurrent precursor domain is larger than what is required to achieve statistically steady turbulent statistics, we spin-up turbulence on an off-line precursor that runs alone on a 6 km × 3 km × 1 km domain for $120,000$ s. During this phase, inflow slices are saved and used to spin up the flow in the concurrent precursor domain, which requires at this point just a single flow through time since the inflow already contains fully developed turbulence. When the domain is filled by resolved turbulence (i.e. after a bit more than one flow through time) boundary conditions in the concurrent precursor are switched to periodic and there is no need for the inflow database anymore. In fact, the concurrent precursor would proceed like a conventional precursor. This procedure is detailed in Stipa et al. (2023).

Table 4: delta h not defined. I can see it in the text in the next paragraph, but I think it should be defined in the paragraph above, where the other parameters are described.

This has been added.

532: 'wake models alone': do you mean 'engineering wake models alone'?
'would predict very similar power production for each individual turbine in the two cases': Should you show the resulting TI values too to be able to state this? I suspect that the resulting TI at HH is similar for both N1 and N2, but may be this should be stated/checked.

We included the reviewer suggestions. Indeed the hub-height TI is the same for the two cases and mention has been added.

Table 5: what is qmin?

This is mentioned in the table caption, it is the minimum heat flux within the boundary layer and it is a useful parameter if these cases will be compared against other codes or ABL conditions in the future.

Should the Froude numbers be added to this table for quick recollection of which case is which??

Good suggestion. We have added this to the caption.

594-607: It would be really interesting to develop this section a bit, by adding some plots illustrating what you describe. Since the way engineering models capture (or not) the pattern of production down a line of turbine, and how accounting for blockage might change the redistribution of power between the front and the back of the wind farm are both still hotly debated in the industry.

We expanded the section following the reviewer's suggestions, adding a plot that shows the effect of mirroring/no mirroring on row-averaged power distributions for the MSC and wake model.

The LES, by resolving the direction fluctuation, will naturally include some effect that would represent wake meandering, while the engineering models, when operating steady state, tend to ignore this. Did you account for direction uncertainty when processing the results from the engineering models? Something like the averaging discussed in Gaumond et al ,Wind Energ. 2014; 17:1169–1178?

Same question about the results from the MSC? Are the results for a single wind direction aligned with the wind farm layout, or are you also averaging results for a finite sector width, accounting for direction uncertainty?

No, all engineering models (MSC, original 3LM, wake model with and without local induction) are run on a single wind direction without accounting for direction uncertainty.

601: 'in agreement with many previous literature studies' . Please include references.

References added as per reviewer's suggestion.

609: 'As the NREL 5-MW thrust curve is not available in official literature..': this must have been an issue for the LES run too no? How did you deal with this then? Should this be mentioned earlier, when mentioning NREL turbine used in the LES? was the thrust and PC used in the LES consistent with that used in the MSC model?

The LES does not require $C_T$ as an input. Instead, our actuator model implementation features blade information and wind turbine controllers. Each radius station is characterized by chord, twist and airfoil lift and drag coefficients, so that $C_T$ is actually an output of our simulations. For this reason, we decided to run isolated turbine simulations extracting the thrusts and power curves with TOSCA. This aspect is highlighted in Appendix B.

610: 'with uniform, non-turbulent inflow': non-turbulent ? really ? thrust and PC curve typically depend on background conditions, such as TI, shear, ... is this the right approach? Or when saying 'non-turbulent', do you mean the turbine is operating in background turbulence only (i.e. no added turbulence from neighbouring turbines).

The turbine curves were extracted using an idealized setup, i.e. not including ground effects, turbulence and shear. We agree with the reviewer that this might not be the ideal method, but the inclusion of these effects would have been extremely computational intense, as a precursor simulation for each set-point would have been required. Moreover, according to Bardal and Saetran (2017), the effect of TI is minimal around the operating conditions chosen in our paper for the wind turbines, i.e. far from cut-in and rated wind speeds (where TI effects are more pronounced). Regarding shear, we did not include its effect in the power curve, but instead shear was considered in the velocity profile from which the background velocity at turbine quadrature points is evaluated. In fact, modeling each rotor as a group of quadrature points allows to account for non-uniform flow conditions (partial waking and shear) at turbine locations.

639: 'while depth averaged perturbation velocities are overestimated ': similar to earlier comment: It would be interesting to find out how your MSC results would change if you work out what is the typical wind direction standard deviation at one point in the precursor LES run, use this as a measured of the wind direction (WD) uncertainty and carry out additional MSC runs for directions 1-2 stdev away, then average the MSC results over a few directions. My expectations would be that the pressure signal is not changing much but the velocity might.

This is an interesting test to perform. In general, we believe that a slightly different wind direction has a greater impact at the microscale rather than on the mesoscale. Focusing on the latter, we agree with the reviewer that, while the 3LM pressure might not change much, the 3LM perturbation velocity could. However, only 3LM pressure is then used in the 3LMR (reconstruction step) to reconstruct the background perturbation velocity from the pressure field. As a consequence, we expect the overall sensitivity of the mesoscale sub-model on small changes in wind direction to be negligible compared to the one of the wake model, as this might enable partial waking, thus appreciable variation in power distributions.

644: 'While such limitations...' : Surely the pressure field is a function of the velocity deficit within the wind farm (which via mass conservation which conditions the vertical displacement at the inversion, which itself will feedback on the pressure). So, is the fact that different velocity distributions between the LES and MSC lead to similar pressure distributions a happy accident? i.e. does it relate to the integrated velocity deficit within the wind farm rather than the shape of the velocity deficit profile?

Looking at the 3LM equations, it can be seen that, once the pressure perturbation is known, the boundary layer displacement is automatically known from linear theory. Moreover, mass conservation can be rewritten for the variable $\eta = \eta_1 + \eta_2$, which depends on the perturbation velocity in both the wind farm and upper layer. Looking at Fig. 2, it can be seen that the error on perturbation velocity w.r.t the LES is opposite in the wind farm and upper layer, i.e. where deficit is overestimated in one it is underestimated in the second.

[Figure]

**Figure 2.** Perturbation pressure (left) and perturbation velocity in the wind farm (center) and upper layers (right). Both cases N1 (subcritical) and N2 (supercritical) are shown. Solid lines indicate LES results, while dashed lines indicate the 3LM results.

.

More than an happy accident, this is a structural problem of the 3LM, and arises because the layer thickness is constant throughout the wind farm. In reality, the wind farm layer grows until a fully developed state or the wind farm end are reached. Conversely, as this is constant in the model, wake effects are stronger towards the end (less height to which deficit is distributed) and smaller at the wind farm entrance (height over which deficit is distributed is larger than actual IBL height).

Figure 11: LES results are time averaged? over what time period? Surprised by the streakiness along the flow direction. What is causing this? not enough distance between the periodic inflow/outflow, leading to turbulence structures which have wrong spatial correlation properties?

Successor runs are carried out for 30,000 s, of which 5,000 s are of spin-up (slightly more than one successor flow turnover time) and 25,000 s are used to perform averaging (Sec. 3.1). The appearance of the streaks suggests that this may not be sufficient to obtain perfectly clean average fields, and the effect of turbulence may still be observed. As a reference, we report below a comparison between two hub-height

averages obtained from two similar cases where one has been run for 40000 s, while the second has been run for 20000 s with spanwise shift of the inflow condition. This procedure is used for faster convergence of the average fields, as streaks do not remain locked in position at the hub height, where the flow is aligned with the x direction, but are slowly shifted along the spanwise direction with a given shift velocity (this velocity is not applied to the flow but physically to the inflow data, so that the flow is still aligned with the x direction even with shifting). As can be seen, while high-frequency oscillations can be detected in the first case, the second simulation depicts almost a perfectly converged average velocity despite running for half the time. We do not believe that such shortcoming causes any problem apart from visualizations of lower quality. In fact, it can be removed completely by increasing the time over which averaging is performed. We believe that the obtained averages have converged sufficiently for the purpose of the present study, and an increased averaging time would have represented an unjustified use of additional computational resources.

[Figure]

**(a)**

**(b)**

**Figure 3.** (a) Hub-height wind speed for the simulation that ran for 40000 s without spanwise inflow shift; (b) Hub-height wind speed for the simulation that ran for 20000 s with spanwise inflow shift.

720: 'at least 10%' . Please clarify that this is for the simulated conditions, at high thrust.

This has been added.

731: again, please clarify that the lapse rate is above the boundary layer.

This has been added.

732: mention of inversion strength, but not inversion thickness. Is this because your results are not sensitive to this?

We did not investigate the sensitivity on inversion thickness. However, the capping inversion is modeled as a temperature jump with zero thickness in the MSC model. Hence, this parameter is only an input for the LES simulations. With increasing inversion thickness, the temperature profile would be more and more smoothed across the inversion, leading to something that is not strictly an inversion anymore, but rather a variable stratification in the free atmosphere. Extension of the 3LM to an arbitrarily stratified free atmosphere has been covered by Devesse et al. (2022).

**References**

Adams, N., Rodaway, C., Gottschall, J., Hawkes, G., and Simon, E.: WESC 2023 Mini-Symposium - OWA GloBE: Building Industry Consensus on the Global Blockage Effect in Offshore Wind, https://doi.org/10.5281/zenodo.8085205, 2023.

Allaerts, D. and Meyers, J.: Boundary-layer development and gravity waves in conventionally neutral wind farms, Journal of Fluid Mechanics, 814, 95–130, https://doi.org/10.1017/jfm.2017.11, 2017.

Allaerts, D. and Meyers, J.: Gravity Waves and Wind-Farm Efficiency in Neutral and Stable Conditions, Boundary-Layer Meteorology, 166, https://doi.org/10.1007/s10546-017-0307-5, 2018.

Allaerts, D. and Meyers, J.: Sensitivity and feedback of wind-farm-induced gravity waves, Journal of Fluid Mechanics, 862, 990–1028, https://doi.org/10.1017/jfm.2018.969, 2019.

Allaerts, D., Broucke, S. V., van Lipzig, N., and Meyers, J.: Annual impact of wind-farm gravity waves on the Belgian-Dutch offshore wind-farm cluster, 2018.

Bardal, L. M. and Saetran, L. R.: Influence of turbulence intensity on wind turbine power curves, Energy Procedia, 137, 553–558, https://doi.org/https://doi.org/10.1016/j.egypro.2017.10.384, 14th Deep Sea Offshore Wind R&D Conference, EERA DeepWind, 2017.

Calaf, M., Meneveau, C., and Meyers, J.: Large eddy simulations of fully developed wind-turbine array boundary layers, Physics of Fluids, 22, https://doi.org/10.1063/1.3291077, 2010.

Churchfield, M., Lee, S., Moriarty, P., Martínez Tossas, L., Leonardi, S., Vijayakumar, G., and Brasseur, J.: A Large-Eddy Simulation of Wind-Plant Aerodynamics, https://doi.org/10.2514/6.2012-537, 2012.

Devesse, K., Lanzilao, L., Jamaer, S., Lipzig, N., and Meyers, J.: Including realistic upper atmospheres in a wind-farm gravity-wave model, Wind Energy Science, 7, 1367–1382, https://doi.org/10.5194/wes-7-1367-2022, 2022.

Lanzilao, L. and Meyers, J.: An Improved Fringe-Region Technique for the Representation of Gravity Waves in Large Eddy Simulation with Application to Wind Farms, Boundary-Layer Meteorology, https://doi.org/10.1007/s10546-022-00772-z, 2022.

Lanzilao, L. and Meyers, J.: A parametric large-eddy simulation study of wind-farm blockage and gravity waves in conventionally neutral boundary layers, 2023.

Nieuwstadt, F. T. M.: On the solution of the stationary, baroclinic Ekman-layer equations with a finite boundary-layer height, Boundary-Layer Meteorology, 26, 377–390, https://doi.org/10.1007/BF00119534, 1983.

Rampanelli, G. and Zardi, D.: A Method to Determine the Capping Inversion of the Convective Boundary Layer, Journal of Applied Meteorology, 43, 925 – 933, https://doi.org/10.1175/1520-0450(2004)043<0925:AMTDTC>2.0.CO;2, 2004.

Smith, R. B.: Gravity wave effects on wind farm efficiency, Wind Energy, 13, 449–458, https://doi.org/https://doi.org/10.1002/we.366, 2010.

Stipa, S., Ajay, A., Allaerts, D., and Brinkerhoff, J.: TOSCA - An Open-Source Finite-Volume LES Environment for Wind Farm Flows, Wind Energy Science, 2023.

---

## Author Comment (AC2)

**University of British Columbia**

**UBCO-UL NSERC Alliance Grant "Reduced-Order Models of Wind Farm Induction and Far-Field Wake Recovery"**

**Response to Reviewer 1**

Exec. S. Stipa - November 14, 2023

We would like to thank the reviewer for the time dedicated to revising the paper. We proceed with answering and clarifying, where possible, the proposed comments.

Our response, denoted in black, is shown below. Modified text of the paper is shown between quotes in *italic*, while the reviewers' comments are denoted in blue. Please refer to the track changes section at the end of this document for a detailed overview of the changes made to the manuscript.

Excellent work describing the MSC wind farm engineering model framework to simulate array interaction effects through a systematic approach that separates mesoscale and microscale scales. The authors made a great job breaking down the model in its different components and analyzing the differences with respect to simpler (industry standard) wake models and verifying against benchmark results from a high-fidelity LES model. The methodology is sound and the results speak for themselves as to the significant improvements from the earlier 3LM model.

I only have a few editorial suggestions and a couple of discussion points around the tuning and the practical application of the model.

In particular, is the re-tuning of the TI model not a contradiction with the modularity principle discussed in section 2.2 where one can "build upon already-existing sub-models, so that additional tuning parameters or individual sub-model re-tuning are not required". I wonder if the tuning coefficients for a stand-alone wake model are equivalent to those used in the MSC framework. Is it appropriate to tune the TI model against LES simulations that account for global blockage effects while the wake model doesn't? Is the separation of scales not happening in TI for now but something worth exploring in the future?

We were surprised to notice how big of an impact the TI model – which basically sets the wake expansion coefficient in the wake model – has on the individual power of each wind turbine in the farm. We believe – as mentioned in the paper – that such strong impact is mainly related to our choice of simulating an aligned layout, combined with the fact that we are at the TI model validity bounds. Moreover, besides the large variation in turbine power obtained when adjusting turbulence intensity, we also noticed that the TI model itself was underestimating turbulent intensity at the turbine locations when compared against LES data.

Our reasoning is that, without TI model tuning, wake model results are affected by an intrinsic error that is not actually due to the wake model itself, but rather to an erroneous wake expansion produced by an incorrect estimation of the TI at the turbine locations. This holds in general and not only when the wake model is used within the MSC framework. Conversely, re-tuning the TI model allows the wake model to operate with a more accurate value of hub-height TI throughout the farm. Fig. 1 shows the row-averaged TI predicted by the original and re-tuned TI model, as well as the one calculated from LES simulations, where N1 and N2 refer to the subcritical and supercritical conditions, respectively.

Regarding the dependency of TI on free atmosphere stability, we argue that lapse rate and inversion strength do not seem to appreciably affect fluctuations below the CNBL height. Conversely, as background velocity is affected by what happens aloft, we expect the TI to change throughout the wind farm as it is related to the ratio of fluctuations over mean. Despite these considerations, the Niayifar and Porté-Agel (2016) TI model does not include stability dependency and it only depends on pre-existing hub-height TI, turbine characteristics and wind farm layout. Nevertheless, when the model predicts TI values that are closer to the LES, it allows the wake model to accurately capture the mean power distributions. This means that row-to-row variations in turbulence intensity due to free atmosphere stability effects may not be a critical aspect to capture. Notably, we do not consider internal ABL stability in our paper, which is expected to produce a substantial difference on the TI experienced at the waked turbines. However, we expect ABL stability to produce more of a global shift in TI throughout the waked turbines, rather than a relative variation as is the case for free atmosphere stability.

[Figure]

**Figure 1**

To conclude, the tuning of the TI model has been performed in order to remove TI-related error from the wake model results. This is something that would have affected all engineering model runs presented in the paper. The message that we want to convey with the sentence *"build upon already-existing sub-models, so that additional tuning parameters or individual sub-model re-tuning are not required"* is that (1) the MSC model operates each individual model in a way that is consistent with that model's design, and (2) the coupling strategy can not be tweaked, i.e. there are no tuning parameters related to model coupling. This means that if a different sub-model is used instead of the ones proposed (we mainly refer to local-induction, wake and TI models), it does not require re-tuning just because it is now operating in a multiscale framework. Unfortunately, this does not mean that the overall MSC framework is a parameter-free model in its entirety, as each of the sub-model parameters still remain. In this regard, the TI model tuning does not conflict with what is stated above, i.e. the model has been re-tuned just because it was performing poorly for this application, but the adopted value of $d_s$ is the same throughout the paper regardless of the wake model being operated alone or coupled with a mesoscale model.

The authors use LES as "ground truth" to tune the TI model and quantify the correctness of the model in very specific neutral conditions. However, from an application point of view, the coefficients of an engineering model are also means to correct the mean bias when the model is integrated over the annual wind climate in a wide range of layout and siting conditions. I wonder if the authors could provide some insights about the foreseen calibration strategy using (observational) validation data, where separation of scales may not be possible. I guess LES offers a high-fidelity benchmark model to tune engineering models but you still need to train the models with measured data to mitigate outstanding biases. Would this training focus on some of the engineering model coefficients or additional ones to preserve the separation-of-scales principle?

The reviewer points to a very important question related to our paper. The proposed separation of scales principle is used when setting up the foundation of our coupling strategy, i.e. distinguishing between microscale and mesoscale effects. In this regard, individual sub-models should be in principle designed and tuned by only looking at their scale, i.e. without allowing different scales to introduce a bias. Just to give an example, individual wake models should be tuned using isolated turbines.

On the other hand, as the author states, "the coefficients of an engineering model are also means to correct the mean bias when the model is integrated over the annual wind climate in a wide range of layout and siting conditions", which is true and allows to indirectly include physical processes and effects that models

do not directly capture in their formulation. However, depending on the bias that one wishes to correct, such practice may lead to erroneous power results if e.g. tuning is performed on velocity or vice versa. This is expected to hold for any engineering model, being it a simple wake model or the MSC framework, with decreasing error on those variables not used for tuning with an increase in the amount of modeled physics. Regarding the MSC framework, we highlight that the proposed mesoscale sub-model does not feature tuning parameters, and it is just a result of the input background state. For this reason, provided that such state is evaluated with the highest possible degree of accuracy, model tuning can be performed at the microscale, i.e. the scale that most directly influences wind turbines. This practice obviously violates the separation of scale principle, as the quantity used for tuning inevitably contains effects from other scales. In particular, while more accurate predictions on the tuning variable might be obtained, correlation could be lost on some of those variables that are not object of tuning.

On an operational standpoint, tuning should be performed first on those models that are characterized by a single output (when data is available). For example, the TI model can only be tuned on TI, hence it should be tuned first, as the overall model can only benefit from this. Possible decorrelation on secondary outputs arising from this operation should be seen as a structural deficiency of dependent models (e.g. the wake models), and can be corrected with further tuning. For example, correcting TI predictions in our case revealed that the wake model was actually performing fairly well and it did not require any tuning. Regarding local induction, because self-induction is excluded when performing power and thrust interpolation from turbine curves, we demonstrated in our paper that it has a small effect on wind farm power and its internal trends. For this reason, we do not recommend tuning the induction model. Finally, the wake model can be tuned by changing the coefficients of Eqs. 10 and 12 to better match power or velocity but, in either case, it is expected to become less accurate at capturing the wake of an isolated wind turbine.

In conclusion, the separation of scales can be enforced from the input in order to generate the model outputs, but there is no current means of splitting observations (characterized by outputs + unmodeled physics) in their individual scale contributions. For this reason, tuning against observations will likely make individual sub-models less accurate at the scale in which they are defined, but capable of masking their lacked physical modeling when coupled. We have the opinion that tuning should begin with the TI model – if TI data is available – then act on the wake model. However, the beneficial effect gained with tuning is expected to be site and layout specific.

Compared to traditional wake models, the MSC model requires significantly more input quantities, some of which are not accessible in wind resource measurement campaigns. While this may fall outside the scope of the paper, I would recommend the authors to discuss how the model could be used in connection to wind farm design. While the model shows significantly higher physical insight compared to traditional wake models it remains to be seen if the additional complexity does not bring additional uncertainties. Further investigation should focus on evaluating the added value of each module in the framework by quantifying accuracy in relevant quantities like AEP and array efficiency over a wide range of layout/siting conditions. These validation datasets would be useful to investigate training methodologies that provide a good trade-off between physical/numerical complexity and accuracy.

We fully agree with the reviewer. The model requires substantial more inputs due to the broader range of physical processes being modeled. The most important piece of information currently missing in site assessment studies is the vertical profile of potential temperature. This provides plenty of important information such as ABL height, ABL and free atmosphere stability. Reanalysis data, such as ERA5, have already been used to address this shortcoming (Allaerts et al., 2018), but it remains to be seen how accurate these annualized energy production (AEP) predictions are when compared with real observations.

We are currently making efforts in this direction. The MSC model has been implemented into OpenWind®

and is currently being tested against observations gathered from the GloBE project (Adams et al., 2023). This will be a step in the direction of better understanding each sub-model's contribution under different conditions.

Regarding wind farm design, thermal stratification and ABL height statistics for the site of interest are essential to produce a representative picture of what the beneficial/detrimental effects of blockage and gravity waves will be. If one does not possess such information, blockage and gains/losses associated with gravity wave effects cannot be predicted. If the amount of required additional statistics is reduced to solely the ABL height (i.e. no info on stratification), global blockage can still be predicted to a certain degree by using the rigid-lid approximation (Smith, 2023). This means that only the flow confinement under the inversion layer is considered, while its deformation is assumed zero for all conditions. While this approach is better than modeling local blockage only, as global blockage associated with flow confinement is included, it completely neglects gains/losses associated to gravity wave effects.

53: duplicated "the".

corrected.

109: For brevity, I would avoid the outline of the section "The present section is organized..." (likewise in subsequent sections).

Thanks for this suggestion. Nonetheless, given the consistent length of the paper, we would like to maintain those outlines to help provide a structured layout of the paper.

185: what about Ct? which wind speed is it based upon?

We rephrased this paragraph, specifying how $C_T$ is calculated for an isolated wind turbine. For waked conditions, we cross-reference Sec. 2.2.4, where we explain how the "freestream" velocity is evaluated in this case.

207: the re-tuning of $d_s$ introduces quite a significant change in the turbulence intensity parameterization. This is indicative of the potentially large dependencies on the layout characteristics. Wouldn't this imply that we need to recalibrate the model with additional LES simulations in new layouts? Otherwise the potential benefits from adding a better description of the mesoscale conditions would be compromised by the underlaying uncertainty of the microscale wake model. The additional cost of this recalibration would also penalize the use of the model as an engineering model.

The turbulence intensity at turbine locations, as explained in the first answer, has a huge effect on the wake model results. However, as shown in the paper, its lack of dependency on free atmosphere stratification does not impair the accuracy of power predictions. For this reason, while the re-calibration of the TI model certainly improves turbine results from an absolute standpoint, this is something mainly related to the microscale and has little effect on the mesoscale. Regarding the latter aspect, i.e. the effect of TI on the mesoscale solution and thereby its indirect influence on the wake model results, the following has to be considered. While the actual magnitude of the Gaussian-filtered wind farm thrust distribution is calculated with the wake model, hence it depends on TI, its shape is mainly influenced by wind turbine layout and potential temperature structure, which do not depend on TI.

[Figure]

**(a)** un-tuned $d_s$        **(b)** tuned $d_s$

**Figure 2.** Differences in inversion layer displacement between tuned and un-tuned $d_s$ coefficient.

In this regard, it can be shown that the difference in magnitude produced using the tuned or un-tuned TI model leads to negligible changes in the mesoscale perturbation solution (illustrated in Fig. 2). As stated above, this is because the TI model does not affect the shape of the forcing, while its value only changes by a small amount w.r.t its order of magnitude. Conversely, at the microscale level, this fraction is already considered important from a power production perspective, as it is around 8% of the total wind farm power. The effect of tuning the TI model is shown in Fig. 3, which highlights how using an un-tuned TI model yields results that are comparably compromised in both wakes-only + induction and coupled modes of operation. Note that, as the error due to blockage is not relevant in this context, results are normalized with the first row-power. Moreover, regarding the coupled model, we only performed one iteration, meaning that turbine thrust distribution is only updated once based on mesoscale model results.

[Figure]

**Figure 3.** Row-averaged turbine power for the tuned and un-tuned TI model cases. Computations are shown for the coupled and wakes-only + induction modes of operation, where the former is run in direct mode, i.e. with only one coupling iteration.

In summary, it can be stated that the higher degree of physical description provided by the MSC model is weakly dependent on TI model tuning. Conversely, independent of the adopted modeling technique – coupled or wakes-only approaches – TI model tuning is strongly recommended as it allows to substantially improve predictions of relative trends in wind farm thrust and power throughout the wind farm.

311: "wake models are usually tuned on velocity for an isolated wind turbine wake". Is this statement referring to equation (7) on the definition of the wake velocity deficit with respect to $U_\infty$ (without blockage)? If so, I would provide the cross-reference. Still, I would say the models are "defined", not "tuned", in terms of $U_\infty$ to be consistent with the same definition used in theoretical power/Ct curves for an isolated turbine.

This is exactly what we wanted to convey. We rephrased the sentence according to the reviewer's suggestions.

 "turbine thrust distribution" add cross-referenced equation (18).

Added.

342: I wouldn't call (23) an ABL profile. This is rather a generalization of Monin-Obukhov surface-layer model to account for stability. An ABL model would be valid across all three layers. Surface-layer context is reinforced when you assume the the friction velocity only varies horizontally. Since this is only applied to the first layer I think it is appropriate to call it a surface-layer model. However, when extending the model to simulate stable conditions it may be worth exploring the use of a single-column ABL model with mixing-length parameterization.

We rephrased the sentence according to the reviewer's suggestions.

452: "Reference potential temperature, inversion jump and lapse rate can be prescribed based on observations". These quantities are not measured in conventional wind resource campaigns. Are you suggesting that the model requires these additional measurements? Wouldn't it be more practical to rely on mesoscale simulations to characterize these inputs?

Yes, in order to exploit the model capability those measurements should be gathered, which is not excluded for future campaigns. Mesoscale simulations, as suggested by the reviewer, may represent a more practical way to characterize these inputs. A third option is to use reanalysis data such as ERA5, which we started to do recently following the approach of Allaerts et al. (2018). The sentence has been expanded to clarify these concepts.

473: can you explain what the "fringe region" does?

The fringe region is basically a Rayleigh damping layer where the desired velocity that the flow tries to achieve is temporally and spatially varying. This means that such approach requires a time and spatially resolved inflow field that is equal or larger than the fringe region in order to have knowledge about the desired velocity to apply. This technique is the preferred way in which an user-defined inflow is prescribed in pseudo-spectral codes, as they can only use periodic boundary conditions in the horizontal directions. For TOSCA, which is a finite-volume code, inlet-outlet boundary conditions can be applied, but the fringe region was chosen as it has the added capability of damping the reflections of gravity waves that are generated by the wind farm. The actual fringe region method adopted in TOSCA is thoroughly explained in Stipa et al. (2023) (WES preprint). Other works that employ this technique are, among others, Allaerts and Meyers (2015, 2017); Lanzilao and Meyers (2022); Wu and Porté-Agel (2017).

580 (and elsewhere): Consider using the term "verification" instead of "validation" since you are doing code-to-code comparisons instead of comparing with measurements.

The suggestion has been implemented in the revised paper.

**References**

Adams, N., Rodaway, C., Gottschall, J., Hawkes, G., and Simon, E.: WESC 2023 Mini-Symposium - OWA GloBE: Building Industry Consensus on the Global Blockage Effect in Offshore Wind, https://doi.org/10.5281/zenodo.8085205, 2023.

Allaerts, D. and Meyers, J.: Large eddy simulation of a large wind-turbine array in a conventionally neutral atmospheric boundary layer, Physics of Fluids, 27, 065 108, https://doi.org/10.1063/1.4922339, 2015.

Allaerts, D. and Meyers, J.: Boundary-layer development and gravity waves in conventionally neutral wind farms, Journal of Fluid Mechanics, 814, 95–130, https://doi.org/10.1017/jfm.2017.11, 2017.

Allaerts, D., Broucke, S. V., van Lipzig, N., and Meyers, J.: Annual impact of wind-farm gravity waves on the Belgian-Dutch offshore wind-farm cluster, 2018.

Lanzilao, L. and Meyers, J.: An Improved Fringe-Region Technique for the Representation of Gravity Waves in Large Eddy Simulation with Application to Wind Farms, Boundary-Layer Meteorology, https://doi.org/10.1007/s10546-022-00772-z, 2022.

Niayifar, A. and Porté-Agel, F.: Analytical Modeling of Wind Farms: A New Approach for Power Prediction, Energies, 9, 741, https://doi.org/10.3390/en9090741, 2016.

Smith, R.: The wind farm pressure field, Wind Energy Science Discussions, 2023, 1–14, https://doi.org/10.5194/wes-2023-56, 2023.

Stipa, S., Ajay, A., Allaerts, D., and Brinkerhoff, J.: TOSCA - An Open-Source Finite-Volume LES Environment for Wind Farm Flows, Wind Energy Science, 2023.

Wu, K. L. and Porté-Agel, F.: Flow Adjustment Inside and Around Large Finite-Size Wind Farms, Energies, 10, 2164, https://doi.org/10.3390/en10122164, 2017.

---

## Referee Report (RR1)

Review of revisions [https://wes.copernicus.org/preprints/wes-2023-75/](https://wes.copernicus.org/preprints/wes-2023-75/)

The Multi-Scale Coupled Model: a New Framework Capturing Wind Farm-Atmosphere Interaction and Global Blockage Effects

Sebastiano Stipa, Arjun Ajay, Dries Allaerts, and Joshua Brinkerhoff

Thank you for responding to my first comments. Generally happy with the corrections, except for some remaining fundamental questions I have below.

When looking at the new Figure 9 and revisions associated with it, I am a bit uncomfortable about the justification of running the wake model with mirroring of the turbines when using it within the MSC, yet needing to run it without mirroring when used outside of the MSC framework. I feel that this raises some fundamental questions and demands some justification based on the physics, and what the mirroring is supposed to represent, rather than, just based on the agreement each sub-option gets wrt the LES.

Did you question this? Do you have some potential explanation?

Also looking at Figure 15 again, I realise that the VC model used to model blockage hardly does anything to the result. And I think this is to be thought about in conjunction with the earlier question.

Both these things brought the following thoughts/doubts:

- I believe that the VC model for blockage, as implemented, does not satisfy mass conservation at the wind farm level. Is that right? I expect that if it did (like for example the RHB with wake expansion does), the model would produce a larger magnitude of blockage and create some background acceleration through the wind farm. The results from e.g. wake + RHB-W would be more different from wake results than your wake + VC currently are. If this were the case, you probably would not mind using it with a wake model also implementing the turbine mirroring, as without it, the combined wake + blockage would overpredict the power at the back of the wind farm. You would perhaps think that it's an issue if the wake model on its own no longer captures the power down the line of turbines when operating standalone. But if the wake model is meant to be used in conjunction with a blockage model, it should be validated against pattern of production when used in conjunction with the blockage model anyway, so this is not necessarily an issue*.  Any thoughts?

- Am I right in thinking that the pressure perturbation derived from the 3LM model is not just the feedback from gravity waves, but indeed also accounts for mass conservation at the wind farm level? After all the ABL displacement which comes in equation 5 is very much the result of mass conservation across the layers. If so, does this raise the question as to whether the MSC framework requires a model for wind farm blockage at all, superposing individual turbine inductions? Would you not be double accounting if you were using the 3LM with a blockage model that does a better job at enforcing mass conservation? In fact, how do the MSC results compare with the LES if you don't used the VC model? Based on the little effect the VC model has when used with the wake model, I suspect that MSC results without it would not change much.

Can you please spare some thoughts about the above and amend the paper with what you conclude on this? The main question to address really is whether the MSC should actually be run with a blockage model at all.

* If a make model is to be used together with a blockage correction model (rather than iteratively coupled with a blockage model superposition induction/enforcing mass conservation at the wind farm level), based on leading row correction applied to the wind farm as a whole, then of course, such a model should still capture the pattern of production when operating on its own.

---

## Author Response (AR2)

**University of British Columbia**

**UBCO-UL NSERC Alliance Grant "Reduced-Order Models of Wind Farm Induction and Far-Field Wake Recovery"**

**Response to Reviewer 2**

Exec. S. Stipa - December 21, 2023

We would like to thank the reviewer for the time dedicated to revising the paper. We proceed with answering and clarifying, where possible, the proposed comments.

Our response, denoted in black, is shown below. Modified text of the paper is shown between quotes in *italic*, while the reviewers' comments are denoted in blue. Please refer to the track changes document for a detailed overview of the additions/deletions made in the revised manuscript.

When looking at the new Figure 9 and revisions associated with it, I am a bit uncomfortable about the justification of running the wake model with mirroring of the turbines when using it within the MSC, yet needing to run it without mirroring when used outside of the MSC framework. I feel that this raises some fundamental questions and demands some justification based on the physics, and what the mirroring is supposed to represent, rather than, just based on the agreement each sub-option gets wrt the LES. Did you question this? Do you have some potential explanation?

We did not include mirroring in the wake model, wake model + VC, and original 3LM model because we wanted to maintain the same wake model configuration as presented in Niayifar and Porté-Agel (2016). Conversely, in the MSC model we added turbine mirroring because we aimed to include the highest amount of physics possible, i.e. we added both the background pressure gradient effect and a more realistic representation of velocity close to the ground through turbine mirroring. The result was that the MSC achieved very good agreement with respect to LES, not only on relative-to-first row, but also on an absolute basis. Regarding the prior models, including turbine mirroring yielded poorer predictions, as they were not defined with this effects included. Nevertheless, it is possible that they might perform better after having re-tuned them to account for turbine mirroring effects. These considerations highlight a confounding artefact of engineering models: when a tuned model is supplemented with additional physics, it may achieve poorer predictive accuracy and it would need to be re-tuned. Specifically, once a model configuration has been selected (i.e. number of rotor quadrature points, superposition method, wake model, mirroring or no mirroring, turbulence intensity model) and tuning is performed, it is very possible that two different configurations may achieve similar results with distinct sets of tuning coefficients.

In our study, we tried to circumvent this artefact by comparing the MSC model to the prior engineering models as they were initially proposed. As we stated in our paper, we do not perform any tuning of the prior engineering models apart from the turbulence intensity model, whose tuning is not aimed at converging turbine power, but rather at providing the correct TI for the TI-wake expansion relation in the wake model (the same coefficients are used for all models). Hence, we can say that in our study the wake model is operated as is, using the same wake superposition technique and TI model used by Niayifar and Porté-Agel (2016) (who also used the Bastankhah and Porté-Agel (2014) model) for the Horns-Rev wind farm. Moreover, Niayifar and Porté-Agel (2016) show the huge difference produced by different superposition strategies or distinct TI models. By looking at Fig. 6 in their paper, it can be seen how energy superposition overestimates power compared to linear. For instance, adding turbine mirroring may change this situation and actually make results better for the energy superposition. Nevertheless, since the set-up used by Niayifar and Porté-Agel (2016) has been proved to work, we kept it the same for our wake model runs. We apply similar reasoning towards the original 3LM and wake model + VC; the first only corrects a first-row bias, while the influence of the second is extremely local, as shown in Fig. 15 of our paper. Basically these model configurations behave like a wake model alone, with some correction, as no meso-scale effects are present. Conversely, as the MSC model brings more physics into the picture, we used mirroring, which is more physical as slower wake growth occurs at the ground. In this case, the longer persistence of the wake is balanced by a favorable pressure gradient inside the wind farm, which is closer to what happens in reality, and we feel makes the MSC model innately more general and less in need of model tuning to cover some missing physics. In fact, no additional tuning was required when the LES featured the same physics that was modeled within the MSC through engineering parametrizations.

*Also looking at Figure 15 again, I realize that the VC model used to model blockage hardly does anything to the result. And I think this is to be thought about in conjunction with the earlier question.*

Regarding this aspect, we would like to give some clarification. While it is true that power and thrust are not much affected by the presence or absence of the VC model (Fig. 15 in our paper), the VC model substantially improves the accuracy of the velocity field (Fig. 13 in our paper). In fact, comparing the first two panels, it can be seen how the velocity depicts a discontinuity at every turbine location without the VC model, while the use of the VC model adds what we refer to as the local blockage, i.e. the induction effect due to the pressure increase in front of each rotor. As explained, since the induction produced by an individual turbine yields a negligible wind speed reduction already five to six diameters upstream of its rotor, local blockage hardly has an effect on power and/or thrust distributions within a wind farm, but its inclusion in the model is important when making velocity estimates in the neighborhood of wind turbines.

*I believe that the VC model for blockage, as implemented, does not satisfy mass conservation at the wind farm level. Is that right? I expect that if it did (like for example the RHB with wake expansion does), the model would produce a larger magnitude of blockage and create some background acceleration through the wind farm. The results from e.g. wake + RHB-W would be more different from wake results than your wake + VC currently are. If this were the case, you probably would not mind using it with a wake model also implementing the turbine mirroring, as without it, the combined wake + blockage would overpredict the power at the back of the wind farm. You would perhaps think that it's an issue if the wake model on its own no longer captures the power down the line of turbines when operating standalone. But if the wake model is meant to be used in conjunction with a blockage model, it should be validated against pattern of production when used in conjunction with the blockage model anyway, so this is not necessarily an issue\*. Any thoughts?*

The reviewer is correct that the VC model does not conserve mass. We coupled the wake model and the VC model as explained in Branlard and Meyer Forsting (2020), which does not conserve mass at the wind farm level. This is because wake models are defined on velocity, while the VC model is defined on vorticity. We are not aware of the details of the RHB with wake expansion. To our knowledge, the RHB model has been only published in Gribben and Hawkes (2019), but Nygaard et al. (2020) state, "*Away from the rotor the flow patterns of this potential flow model* [the RHB] *are nearly identical to those of the vortex cylinder*." In general, we do not believe that the cumulative induction from single rotors equates to the large-scale pressure perturbation around the wind farm, as this effect is very local. We agree with the reviewer's considerations about using the RHB model with turbine mirroring, but we emphasize that the local blockage model should only capture local blockage in order to lie within the scales separation principle mentioned in our study. If this is not the case, there might be some bias due to double counting between the local induction model and the mesoscale global blockage model. If the reviewer is referring to the upgraded RHB model mentioned in Nygaard et al. (2022) and developed by the same authors, it is properly a global blockage model, so care should be paid in this case to avoid double counting global blockage effects.

In general, it is extremely difficult to validate each sub-model with observations, as these would contain all the ongoing physics. One alternative is to use computer simulations that incrementally add more physics to the problem. For example, to validate the wake model + VC, one could run LES simulations of a truly neutral (i.e. fully non-stratified) boundary layer, with the domain top placed very far from the wind farm in order to avoid any flow confinement (which would produce large-scale pressure gradients). In this case, as there are no gravity wave induced pressure perturbation, the mesoscale model can be disregarded and the accuracy of the turbine-scale parametrization can be assessed. A simulation of this type has been performed in Lanzilao and Meyers (2023) and it shows that, without thermal stratification, pressure perturbations only exist very close to the wind turbines (Fig. 15 in Lanzilao and Meyers, 2023). Moreover, looking at Fig. 21 in their paper, it can be seen how the pattern of production for the NBL (neutral boundary layer) case is

very similar to the wake model with turbine mirroring (Fig. 9 in our study). In our paper, the wake model without mirroring matches better with our simulations, but these include stratification. So basically it is as if not including mirroring reduces the error on the pattern of production produced by stratification effects, where the favorable pressure gradient within the farm produces a faster wake recovery. Finally, blockage in truly neutral conditions is only produced by the superposition of individual turbine inductions (i.e. local blockage effects), while boundary layer displacement does not produce any pressure disturbance in this case due to the absence of stability.

To conclude, as proved by Fig. 9 in our paper, the fact that the wake model in the same configuration as the one used in the MSC model (wake model alone with mirroring in our case) does not capture the pattern of production is not an issue as long as this pattern is captured by the overall MSC model. Moreover, our LES simulations contains stratification, while the wake model alone does not. To be strict, the wake model alone is trying to model a wind farm immersed in a fully neutral boundary layer. The fact that it captured the pattern of production is simply a consequence of having used the wake model in a configuration that was shown by Niayifar and Porté-Agel (2016) to be the best for standalone use.

Am I right in thinking that the pressure perturbation derived from the 3LM model is not just the feedback from gravity waves, but indeed also accounts for mass conservation at the wind farm level? After all the ABL displacement which comes in equation 5 is very much the result of mass conservation across the layers. If so, does this raise the question as to whether the MSC framework requires a model for wind farm blockage at all, superposing individual turbine inductions? Would you not be double accounting if you were using the 3LM with a blockage model that does a better job at enforcing mass conservation? In fact, how do the MSC results compare with the LES if you don't used the VC model? Based on the little effect the VC model has when used with the wake model, I suspect that MSC results without it would not change much.

The reviewer is correct that results would not change much by removing the VC model, but this only holds for the wind farm power production and thrust. The velocity would be incorrect by not using the VC model, as the background velocity calculated by the mesoscale model does not include local blockage effects. This can be proved mathematically. First, the 3LM evaluates the inversion layer displacement such that mass conservation is satisfied within each layer. Moreover, the pressure disturbance is the one that puts in agreement the momentum balance below $H$ and the pressure response given by the resulting boundary layer displacement. If a fully neutral flow is considered, the only pressure disturbance in the boundary layer is due to the local induction, but in this case the 3LM model predicts a null pressure disturbance, as $\Phi \rightarrow 0$ in Eq. 5 (this is also shown in Smith, 2023, Tab. 2), hence the 3LM disregards local blockage. In this case, mass imbalance produced by velocity decrease is fully compensated by streamline divergence.

In conclusion, while local blockage does not affect power and thrust distribution, its parametrization using for example the VC or the RHB models should be retained for an accurate velocity field in the region immediately upstream of the wind farm. Moreover, if the local induction model only models local blockage effects, without attempting to cover a broader range of physical aspects, the separation of scales principle holds and global induction is not double counted. Conversely, local induction cannot be double counted as the 3LM neglects it by construction.

Can you please spare some thoughts about the above and amend the paper with what you conclude on this? The main question to address really is whether the MSC should actually be run with a blockage model at all.

The paper explicitly clarifies this aspect at line 151: "*It is important to highlight the significance of the pressure variable, as this concept will be essential in the formulation of the MSC framework later on. First of all, Allaerts and Meyers (2019) make the assumption of zero vertical pressure gradient inside the ABL, hence p only varies in the horizontal directions. Secondly [...] it is clear that p can only change in response*

*to a vertical displacement of the ABL and only if the latter experiences a capping inversion jump $\Delta\theta$ and/or a stable lapse rate $\gamma$. For this reason, the pressure variable considered by Allaerts and Meyers (2019) only contains the effect of internal and interfacial waves, and does not account for the local pressure rise in front of each wind turbine, responsible for what we refer to as local blockage or turbine-level induction, which must be modeled separately.*"

We also added an additional reminder of this aspect in the revised paper at the beginning of Sec. 2.2.4, namely "*As explained at the end of Section 2.1.1, the pressure-induced background velocity produced by the atmospheric perturbation model at the mesoscale does not contain local blockage effects.*"

Finally, we significantly expanded our discussion regarding turbine mirroring at the beginning of Sec. 4. Please refer to the track changes document.

**References**

Allaerts, D. and Meyers, J.: Sensitivity and feedback of wind-farm-induced gravity waves, Journal of Fluid Mechanics, 862, 990–1028, https://doi.org/10.1017/jfm.2018.969, 2019.

Bastankhah, M. and Porté-Agel, F.: A new analytical model for wind-turbine wakes, Renewable Energy, 70, 116–123, https://doi.org/https://doi.org/10.1016/j.renene.2014.01.002, special issue on aerodynamics of offshore wind energy systems and wakes, 2014.

Branlard, E. and Meyer Forsting, A. R.: Assessing the blockage effect of wind turbines and wind farms using an analytical vortex model, Wind Energy, 23, 2068–2086, https://doi.org/https://doi.org/10.1002/we.2546, 2020.

Gribben, B. J. and Hawkes, G. S.: A potential flow model for wind turbine induction and wind farm blockage, Systems and Engineering Technology, 2019.

Lanzilao, L. and Meyers, J.: A parametric large-eddy simulation study of wind-farm blockage and gravity waves in conventionally neutral boundary layers, 2023.

Niayifar, A. and Porté-Agel, F.: Analytical Modeling of Wind Farms: A New Approach for Power Prediction, Energies, 9, 741, https://doi.org/10.3390/en9090741, 2016.

Nygaard, N. G., Steen, S., Poulsen, L., and Pedersen, J. G.: Modelling cluster wakes and wind farm blockage, Journal of Physics: Conference Series, 1618, 062 072, https://doi.org/10.1088/1742-6596/1618/6/062072, 2020.

Nygaard, N. G., Poulsen, L., Svensson, E., and Grønnegaard Pedersen, J.: Large-scale benchmarking of wake models for offshore wind farms, vol. 2265, p. 022008, https://doi.org/10.1088/1742-6596/2265/2/022008, 2022.

Smith, R.: The wind farm pressure field, Wind Energy Science Discussions, 2023, 1–14, https://doi.org/10.5194/wes-2023-56, 2023.